



# Impacts of Ice-Particle Size Distribution Shape Parameter on Climate Simulations with the Community Atmosphere Model Version 6 (CAM6)

Wentao Zhang[1], Xiangjun Shi[1,*], and Chunsong Lu[2]

[1] School of Atmospheric Sciences, Nanjing University of Information Science and Technology, Nanjing 210044, China.

[2] Key Laboratory for Aerosol-Cloud-Precipitation of China Meteorological Administration, Nanjing University of Information Science and Technology, Nanjing 210044, China.

*Correspondence to*: Xiangjun Shi (shixj@nuist.edu.cn)

**Abstract.** The impacts of the ice crystal size distribution shape parameter ($\mu_i$) were considered in the two-moment bulk cloud microphysics scheme of the Community Atmosphere Model Version 6 (CAM6). The calculating formulas of statistical

mean radii indicate that, under the same mass ($q_i$) and number ($N_i$) of ice crystals, the ratios of the mass-weighted radius ($R_{qi}$, not related to $\mu_i$) to other statistical mean radii (e.g., effective radiative radius) are completely determined by $\mu_i$. Off-line tests show that $\mu_i$ has a significant impact on the cloud microphysical processes owing to the $\mu_i$-induced changes in ice crystal size distribution and statistical mean radii (excluding $R_{qi}$). Climate simulations show that increasing $\mu_i$ would lead to higher $q_i$ and lower $N_i$ in most regions, and these impacts can be explained by the changes in cloud microphysical processes. After

increasing $\mu_i$ from 0 to 5, the longwave cloud radiative effect increases (stronger warming effect) by 5.58 W m$^{-2}$ (25.11%), and the convective precipitation rate decreases by $-0.12$ mm day$^{-1}$ (7.64%). In short, the impacts of $\mu_i$ on climate simulations are significant and the main influence mechanisms are also clear. This suggests that the $\mu_i$-related processes deserve to be parameterized in a more realistic manner.

## 1 Introduction

Clouds are an integral part of the Earth's radiation budget and global water cycle (Liou, 1986; Luo and Rossow, 2004; Bony et al., 2015; Zhou et al., 2016). Since cloud microphysical processes occur at scales that are much smaller than the resolution of atmospheric models, it remains a significant challenge for atmospheric models to represent cloud-related processes, especially ice-phase cloud microphysical processes (Spichtinger and Gierens, 2009; Erfani and Mitchell, 2016; Paukert et al., 2019; Morrison et al., 2020; Proske et al., 2021). Because it is impossible for atmospheric models to individually describe

cloud particles (e.g., cloud droplets or ice crystals), only the macrostatistical features of cloud particles are represented in cloud microphysics schemes. From the outset, the development of cloud microphysics schemes has resulted in two distinct categories: bulk microphysics parameterization and spectral (bin) microphysics (Milbrandt and Yau, 2005; Khain et al., 2015). The spectral (bin) approach represents the cloud particle size distributions (PSDs) by using tens to hundreds of bins.





The computational cost of this approach is very high because of the massive interactions among different bins. The bulk
microphysics scheme represents the PSDs by a semiempirical distribution function. Compared to the spectral (bin) scheme,
the bulk microphysics scheme has high computational efficiency and has been widely used in climate models (Morrison et
al., 2005; Lohmann et al., 2007; Salzmann et al., 2010; Gettelman and Morrison, 2015).

In climate models with bulk cloud microphysics scheme, the PSD is usually described by the gamma distribution function
with three parameters, namely, the intercept parameter ($N_0$), the slope parameter ($\lambda$), and the spectral shape parameter ($\mu$)
(Khain et al., 2015; Morrison et al., 2020). Note that the commonly used two-moment bulk microphysics scheme predicts
only the mass and number of cloud particles, which cannot constrain these three parameters (i.e., $N_0$, $\lambda$, and $\mu$). Therefore,
one of these three parameters (typically $\mu$) must be determined from an empirical formula or set to a given value (e.g.,
Morrison and Gettelman, 2008; Barahona et al., 2014; Eidhammer et al., 2017). For instance, the $\mu$ ($\mu_i$) of ice crystals (ICs)
in the two-moment bulk stratiform cloud microphysics scheme developed by Morrison and Gettelman 2008 (hereafter MG
scheme) is set to zero (i.e., the $\mu_i$ is ignored). In recent years, off-line tests and short-term simulations (a few days or less)
with high-resolution atmospheric models (e.g., cloud-resolving models and mesoscale models) have shown that $\mu_i$ has a
significant impact on cloud microphysical processes and synoptic systems (Milbrandt and Yau, 2005; Milbrandt and
McTaggart-Cowan, 2010; Loftus et al., 2014; Khain et al., 2015; Milbrandt et al., 2021). Unlike short-term simulations,
climate simulations pay more attention to the equilibrium states or quasi-equilibrium states because the feedback processes
become important (Sherwood et al., 2015; King et al., 2020). However, in terms of climate simulations, few studies have
focused on the influence of $\mu_i$.

In this study, in order to investigate the impacts of $\mu_i$ on climate simulations with the Common Atmosphere Model version 6
(CAM6) model, the impacts of $\mu_i$ were considered in the MG scheme by a tunable parameter. There were two major
motivations behind this work. First, are the impacts of $\mu_i$ notable? If yes, it's necessary for climate models to represent the $\mu_i$
and $\mu_i$-related processes in a more realistic manner. And second, what are the main mechanisms for these impacts? These
would be helpful to understand the climate simulations with the impacts of $\mu_i$. This paper is organized as follows: the
modified MG scheme and experimental setup are described in Section 2; cloud microphysical process off-line tests and
CAM6 model simulation results are analyzed in Section 3; and finally, the summary and conclusions are provided in Section
4.

## 2 Model and experiments

### 2.1 The modified MG scheme

The CAM6 model, which is the atmospheric component of the Community Earth System Model Version 2.1.3 (CAM6,
Bogenschutz et al., 2018; CESM2, Danabasoglu et al., 2020), was used in this study. It is noteworthy that the treatments of
clouds in climate models are usually divided into two categories: convective cloud schemes with simplified cloud





microphysics and stratiform cloud schemes with relatively detailed cloud microphysics. In the CAM6 model, the convective cloud scheme does not consider the PSD of ICs (Zhang and McFarlane, 1995; Zhang et al., 1998; Bogenschutz et al., 2013; Larson, 2017). The stratiform cloud microphysics was represented by the updated MG scheme with prognostic precipitation (Gettelman and Morrison, 2015). In both versions of the MG scheme, the ICs are assumed to be spherical, and the PSD of ICs is described by the gamma distribution function:

$$N_i(D) = N_{0i}D^{\mu_i}e^{-\lambda_i D} \tag{1}$$

where $N_i(D)$ is the number density of the ICs with diameter D. $N_{0i}$, $\lambda_i$, and $\mu_i$ (nonnegative values) are the intercept parameter, the slope parameter, and the spectral shape parameter, respectively. Given that $\mu_i$ is known, $N_{0i}$ and $\lambda_i$ can be determined by the local in-cloud IC mass mixing ratio and number concentration ($q_i$ and $N_i$, prognostic variables).

$$\lambda_i = [\frac{\pi\rho_i}{6}\frac{N_i}{q_i}\frac{\Gamma(4+\mu_i)}{\Gamma(1+\mu_i)}]^{1/3} \tag{2}$$

$$N_{0i} = \frac{N_i\lambda_i^{(1+\mu_i)}}{\Gamma(1+\mu_i)} \tag{3}$$

where the IC bulk density ($\rho_i$) is 500 kg m$^{-3}$ and $\mu_i$ is zero (i.e., not considered) in the default MG scheme. $\Gamma(x) = \int_0^\infty t^{x-1}e^{-t}\,dt$ is the gamma function. It is noteworthy that the $k$th moment of this size distribution ($M_k$) is found by

integrating the distribution in this form: $M_k = \int_0^\infty N_{0i}D^{\mu_i+k}e^{-\lambda_i D}\,dD = N_{0i}\Gamma(k+\mu_i+1)/\lambda_i^{(k+\mu_i+1)}$ (Eidhammer et al., 2014). Furthermore, the recursive property of the gamma function (i.e., $\Gamma(x+1) = x\Gamma(x)$ ) is also used for the following formula derivation.

Eq. (2) and (3) also indicate that, under the same $q_i$ and $N_i$, changes in $\mu_i$ could impact the other two parameters regarding the PSD of ICs (i.e., $N_{0i}$ and $\lambda_i$). Meanwhile, the number-weighted radius ($R_{ni}$) related to the IC deposition/sublimation process,

the effective radiative radius ($R_{ei}$) used for the radiative transfer scheme, and other statistical mean radii might be influenced. To better understand the influence of $\mu_i$ on the ice-phase cloud microphysical processes, the equations for calculating the statistical mean radii are introduced first. The mass-weighted radius ($R_{qi}$) is calculated from Eq. (4). The number-weighted radius ($R_{ni}$), which is the so-called mathematical mean value, is calculated from Eq. (5). The area-weighted radius ($R_{ai}$) is calculated from Eq. (6). $R_{ei}$, which is defined as the cross-section weighted radius (Schumann et al., 2011; Wyser, 1998), is

calculated from $R_{qi}^3/R_{ai}^2$ (Eq. 7). Note that $R_{qi}$ can be calculated by $q_i$ and $N_i$ (the last term of Eq. 4, without $\mu_i$), and the other statistical mean radii (e.g., $R_{ni}$, $R_{ai}$, and $R_{ei}$) can be calculated by $R_{qi}$ and $\mu_i$ (Eq. 5-7). In other words, the ratios of the other statistical mean radii (e.g., $R_{ni}$, $R_{ai}$, and $R_{ei}$) to $R_{qi}$ are functions of $\mu_i$. For nonnegative $\mu_i$ values, Eq. (5) and (6) indicate that $R_{ni}$ and $R_{ai}$ are always less than $R_{qi}$. This can be explained by the physical reason that larger ICs contribute more to $R_{qi}$ than to $R_{ni}$ and $R_{ai}$. Similarly, $R_{ei}$ is always greater than $R_{qi}$ (Eq. 7). Furthermore, Eq. (5-7) also indicate that with increasing $\mu_i$, $R_{ni}$,

$R_{ai}$, and $R_{ei}$ approach $R_{qi}$. In Section 3.1, more analyses are provided by off-line tests.





$$R_{qi} = \frac{1}{2} \left[ \frac{\int_0^\infty D^3 N_i(D)\, \mathrm{d}D}{\int_0^\infty N_i(D)\, \mathrm{d}D} \right]^{1/3} = \frac{1}{2\lambda_i} \left[ \frac{\Gamma(\mu_i + 4)}{\Gamma(\mu_i + 1)} \right]^{1/3} = \left( \frac{3}{4\pi\rho_i} \frac{q_i}{N_i} \right)^{1/3} \tag{4}$$

$$R_{ni} = \frac{1}{2} \frac{\int_0^\infty D N_i(D)\, \mathrm{d}D}{\int_0^\infty N_i(D)\, \mathrm{d}D} = \frac{1}{2\lambda_i} \frac{\Gamma(\mu_i + 2)}{\Gamma(\mu_i + 1)} = R_{qi} \left[ \frac{\Gamma(1 + \mu_i)}{\Gamma(4 + \mu_i)} \right]^{\frac{1}{3}} \frac{\Gamma(\mu_i + 2)}{\Gamma(\mu_i + 1)} = R_{qi} \frac{(\mu_i + 1)}{[(\mu_i + 3)(\mu_i + 2)(\mu_i + 1)]^{\frac{1}{3}}} \tag{5}$$

$$R_{ai} = \frac{1}{2} \left[ \frac{\int_0^\infty D^2 N_i(D)\, \mathrm{d}D}{\int_0^\infty N_i(D)\, \mathrm{d}D} \right]^{1/2} = \frac{1}{2\lambda_i} \left[ \frac{\Gamma(\mu_i + 3)}{\Gamma(\mu_i + 1)} \right]^{1/2} = R_{qi} \left[ \frac{\Gamma(1 + \mu_i)}{\Gamma(4 + \mu_i)} \right]^{1/3} \left[ \frac{\Gamma(\mu_i + 3)}{\Gamma(\mu_i + 1)} \right]^{1/2}$$

$$= R_{qi} \frac{[(\mu_i + 2)(\mu_i + 1)]^{\frac{1}{2}}}{[(\mu_i + 3)(\mu_i + 2)(\mu_i + 1)]^{\frac{1}{3}}} \tag{6}$$

$$R_{ei} = \frac{1}{2} \frac{\int_0^\infty D^3 N_i(D)\, \mathrm{d}D}{\int_0^\infty D^2 N_i(D)\, \mathrm{d}D} = \frac{(R_{qi})^3}{(R_{ai})^2} = \frac{1}{2\lambda_i} \frac{\Gamma(\mu_i + 4)}{\Gamma(\mu_i + 3)} = R_{qi} \left[ \frac{\Gamma(1 + \mu_i)}{\Gamma(4 + \mu_i)} \right]^{1/3} \frac{\Gamma(\mu_i + 4)}{\Gamma(\mu_i + 3)}$$

$$= R_{qi} \frac{(\mu_i + 3)}{[(\mu_i + 3)(\mu_i + 2)(\mu_i + 1)]^{\frac{1}{3}}} \tag{7}$$

Because $\mu_i$ is zero in the default MG scheme, the equations for the cloud microphysical processes are simplified by omitting $\mu_i$ (Morrison and Gettelman, 2008; Gettelman and Morrison, 2015). In this study, these equations are modified to consider the impact of $\mu_i$ (i.e., nonzero $\mu_i$). In the default MG scheme, there are three cloud microphysical processes, which are related to the PDF of ICs. They consist of the deposition/sublimation of ICs, the autoconversion of IC to snow, and the mass-

weighted and number-weighted IC fall velocities ($V_{qi}$ and $V_{ni}$), respectively. Table 1 shows the original and modified equations for these cloud microphysical processes. The $dq_i/dt$ (i.e., the time derivative of $q_i$) caused by the deposition/sublimation process (including the Wegener-Bergeron process in mixed-phase clouds) is calculated from $dq_i/dt = S_i/(T_p\tau_i)$, where $S_i$, $T_p$, and $\tau_i$ are the ice supersaturation, a psychrometric correction to account for the release of latent heat, and the supersaturation relaxation time scale, respectively (Morrison and Gettelman, 2008). Among them, $\tau_i$ is related to $\mu_i$.

In the original equation of $\tau_i$ (Table 1, left column), $N_{0i} = \lambda_i N_i$ (Eq. 3) and $\lambda_i^{-1} = 2R_{ni}$ at $\mu_i = 0$ (Eq. 5). Therefore, the original equation for $\tau_i$ can be rewritten as the modified equation (Table 1, right column). The modified equation indicates that $\tau_i$ is inversely proportional to $N_i R_{ni}$, which is consistent with the equation obtained by Korolev et al. (2003). This modified equation also indicates that, under the same $q_i$ and $N_i$ ($R_{qi}$ is also fixed), $\mu_i$ can affect $\tau_i$ (i.e., the IC deposition/sublimation process) via the influence on $R_{ni}$. In the MG scheme, ICs with radii greater than the threshold ($R_{cs}$) are considered to be snow.

Correspondingly, the mass and number of ICs converted to snow ($q_{iauto}$ and $N_{iauto}$) are represented by the integration of those ICs with radii greater than $R_{cs}$. Therefore, the incomplete gamma function, ($\Gamma(s, x) = \int_x^\infty t^{s-1} e^{-t}\, \mathrm{d}t$), is used to calculate $q_{iauto}$ and $N_{iauto}$ (right column). It is necessary to note that, at $\mu_i = 0$, the modified equations for $q_{iauto}$ and $N_{iauto}$ can be rewritten as the original equations (i.e., omitting $\mu_i$, left column) based on a property of the incomplete gamma function (i.e.,





$\Gamma(s, x) = (s-1)! \, e^{-x} \sum_{k=0}^{s-1} \frac{x^k}{k!}$, where $s$ is a positive integer). Based on the diameter-fall speed relationship, $V = aD^b$ ($a$ and

$b$ are empirical coefficients), and the properties of the gamma function, $\mu_i$ is considered in the equations for mass-weighted and number-weighted terminal fall speeds ($V_{qi}$ and $V_{ni}$, Table 1).

**Table 1.** Equations for calculating the $\mu_i$-related cloud microphysical processes*.

| | Original ($\mu_i = 0$) | Modification (nonzero $\mu_i$) |
|---|---|---|
| $\tau_i$ | $\dfrac{1}{2\pi\rho_a D_v N_{0i}\lambda_i^{-2}}$ | $\dfrac{1}{2\pi\rho_a D_v N_i 2R_{ni}}$ |
| $N_{iauto}$ | $\dfrac{N_{0i}}{\lambda_i} e^{-\lambda_i 2R_{cs}}$ | $N_{0i}\dfrac{\Gamma(1+\mu_i, \lambda_i 2R_{cs})}{\lambda_i^{1+\mu_i}}$ |
| $q_{iauto}$ | $\dfrac{\pi\rho_i N_{0i}}{6}\left[\dfrac{(2R_{cs})^3}{\lambda_i} + \dfrac{3(2R_{cs})^2}{\lambda_i^2} + \dfrac{6(2R_{cs})}{\lambda_i^3} + \dfrac{6}{\lambda_i^4}\right] e^{-\lambda_i 2R_{cs}}$ | $\dfrac{\pi\rho_i N_{0i}}{6}\dfrac{\Gamma(4+\mu_i, \lambda_i 2R_{cs})}{\lambda_i^{4+\mu_i}}$ |
| $V_{ni}$ | $\dfrac{(\frac{\rho_{a850}}{\rho_a})^{0.35}a\Gamma(1+b)}{\lambda_i^b \Gamma(1)}$ | $\dfrac{(\frac{\rho_{a850}}{\rho_a})^{0.35}a\Gamma(1+b+\mu_i)}{\lambda_i^b \Gamma(1+\mu_i)}$ |
| $V_{qi}$ | $\dfrac{(\frac{\rho_{a850}}{\rho_a})^{0.35}a\Gamma(4+b)}{\lambda_i^b \Gamma(4)}$ | $\dfrac{(\frac{\rho_{a850}}{\rho_a})^{0.35}a\Gamma(4+b+\mu_i)}{\lambda_i^b \Gamma(4+\mu_i)}$ |

*Where $D_v$ is the diffusivity of water vapor in air ($D_v$ is calculated as a function of temperature and pressure, $D_v = 8.794\times10^{-5}\times T^{1.81}/P$); $R_{cs}$ is the threshold radius for the autoconversion of IC to snow ($R_{cs}$ = 100 μm); $\rho_a$ is the air density; $\rho_{a850}$ is the reference air density at 850 hPa,
and $a$ and $b$ are empirical coefficients ($a = 700$ m$^{1-b}$ s$^{-1}$, $b = 1$).

## 2.2 CAM6 experimental design

Observational studies have shown that $\mu_i$ is less than 5 under most conditions (Heymsfield, 2003; McFarquhar et al., 2015). This study focuses only on investigating the influence of $\mu_i$. There are four $\mu_i$-related processes (i.e., the radiative transfer process and three cloud microphysical processes) in the modified CAM6 model. Note that $\mu_i$ can be set to different values for

different processes with the advantage of model simulations. Seven experiments were conducted in this study (Table 2). The Mu0 experiment is considered to be the reference experiment because $\mu_i$ is set to zero for all of the $\mu_i$-related processes. The $\mu_i$ is set to 2 for all of the $\mu_i$-related processes in the Mu2 experiment, and the $\mu_i$ is set to 5 for all of the $\mu_i$-related processes in the Mu5 experiment. The comparison between the Mu2 (or Mu5) and Mu0 experiments shows the influence of $\mu_i$ on climate simulations. Furthermore, to investigate the influence of each $\mu_i$-related process, an additional four experiments, namely,

Tao5, Auto5, Fall5, and Rei5, were conducted. In this study, for ease of expression, "Δ" is used to denote the difference from the Mu0 experiment (e.g., ΔTao5 = Tao5 − Mu0). Without specification, the comparisons between model simulations are relative to the Mu0 experiment. When analyzing a cloud property variable (e.g., $q_i$), it is necessary to know which





experiment the variable comes from. To show this information, the experiment name is added as a superscript. For example, the $q_i$ from the Mu5 experiment is denoted as $q_i^{Mu5}$, the difference in $q_i$ between the Mu5 and Mu0 experiments is denoted as

$q_i^{\Delta Mu5}$, and the relative change of $q_i$ from the Mu5 experiment is denoted as $q_i^{\Delta Mu5/Mu0}$.

In this study, all experiments were atmosphere-only simulations (i.e., sea surface temperature and sea ice are prescribed) with a horizontal resolution of 1.9° latitude × 2.5° longitude and 30 vertical layers. All experiments run for 11 model years and the last 10 years were used for the analyses. In addition, the standard deviation calculated from the averages of each year (i.e., 10 averages) was used to check the statistical significance of the multiyear average (i.e., 10-year average).

**Table 2.** The values of $\mu_i$ in all experiments conducted in this study. $\mu_{i\_tao}$, $\mu_{i\_auto}$, $\mu_{i\_fall}$, and $\mu_{i\_rei}$ indicate the $\mu_i$ used for calculating the IC deposition/sublimation (*tao*), autoconversion of IC to snow (*auto*), IC fall velocity (*fall*), and the $R_{ei}$ used for the radiation scheme (*rei*), respectively.

| Names | $\mu_{i\_tao}$ | $\mu_{i\_auto}$ | $\mu_{i\_fall}$ | $\mu_{i\_rei}$ |
|---|---|---|---|---|
| Mu0 | 0 | 0 | 0 | 0 |
| Mu2 | 2 | 2 | 2 | 2 |
| Mu5 | 5 | 5 | 5 | 5 |
| Tao5 | 5 | 0 | 0 | 0 |
| Auto5 | 0 | 5 | 0 | 0 |
| Fall5 | 0 | 0 | 5 | 0 |
| Rei5 | 0 | 0 | 0 | 5 |

## 3 Results and analysis

### 3.1 Off-line tests

To better understand the impact of $\mu_i$ on climate simulations, the PSD of ICs and $\mu_i$-related cloud microphysical processes are first illustrated by off-line tests. In these off-line tests, the impact of $\mu_i$ was analyzed at a given $R_{qi}$ (i.e., the ratio of $q_i$ to $N_i$ is fixed).

Eq. (1-3) indicate that the normalized IC size distribution (i.e., the relative contributions of each bin) can be calculated from $R_{qi}$ and $\mu_i$. Fig. 1 shows the impact of $\mu_i$ on the PSDs. Under the same $\mu_i$, the shapes of the PSDs (i.e., the relative number or

140 mass contributions of each bin) with $R_{qi} = 20$ µm (small IC scenario) are the same as those with $R_{qi} = 60$ µm (large IC scenario). In other words, the shape of the PSD is completely determined by $\mu_i$ (i.e., spectral shape parameter). As expected, the PSDs move toward larger radii with increasing $R_{qi}$. As introduced in the study of Milbrandt et al. (2021), the PSD becomes narrow with increasing $\mu_i$. Note that, in terms of number, the contributions of the smaller size bins significantly decrease with increasing $\mu_i$. Unlike the number contributions, the mass contributions of the larger size bins significantly

decrease with increasing $\mu_i$ because the mass contribution is more sensitive to the IC radius. Under the large IC scenario (i.e.,



$R_{qi}$ = 60 µm), the mass contribution of the ICs with radii greater than $R_{cs}$ is significantly decreased with increasing $\mu_i$. The above analyses suggest that the cloud microphysical processes that depend on the PSD of ICs (e.g., autoconversion of IC to snow) might be significantly influenced by $\mu_i$.

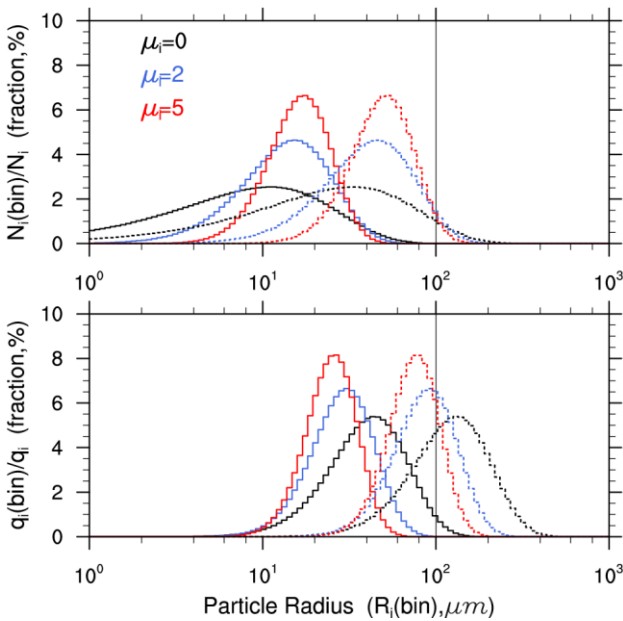

**Figure 1.** The relative number (upper panel) and mass (lower panel) contributions from each radius bin of ICs. Each bin width is the same based on the logarithm of the particle radius. $N_i$ and $q_i$ are the total number and mass of ICs, respectively. A total of 100 bins were used here. The solid lines indicate the normal IC scenario (i.e., $R_{qi}$ = 20 µm), and the dotted lines indicate the large IC scenario (i.e., $R_{qi}$ = 60 µm). The vertical black line indicates the $R_{cs}$ that was used for the autoconversion of IC to snow ($R_{cs}$ = 100 µm).

The off-line tests were performed for the $\mu_i$-related cloud microphysical processes and statistical mean radii (Table 3). As introduced in Section 2.1, $R_{ni}$, $R_{ai}$, and $R_{ei}$ can be calculated from $R_{qi}$ and $\mu_i$. Both $R_{ni}$ and $R_{ai}$ significantly increase with increasing $\mu_i$ (Table 3). This is in agreement with their calculation equations (Eq. 5 and 6). $R_{ni}$ is approximately half of $R_{qi}$ at $\mu_i = 0$ (i.e., 11.00/20 and 33.02/60), while $R_{ni}$ is close to $R_{qi}$ at $\mu_i = 5$ (i.e., 17.26/20 and 51.78/60). According to the calculation equation of $R_{ei}$ (Eq. 7), $R_{ei}$ decreases with increasing $\mu_i$. The ratios of $R_{ei}$ to $R_{qi}$ at $\mu_i = 0$, 2, and 5 are 1.65 (i.e., 33.02/20 and 99.06/60), 1.28 (i.e., 25.54/20 and 76.63/60), and 1.15 (i.e., 23.01/20 and 69.04/60), respectively. It is necessary to point out that with increasing $\mu_i$, both $R_{ni}$, $R_{ai}$, and $R_{ei}$ approach $R_{qi}$ (Table 3) because the PSD of ICs becomes narrow (Fig.1). As expected, $\tau_i$ decreases with increasing $\mu_i$ (Table 3) because $\tau_i$ is inversely proportional to $R_{ni}$ (Table 2). The decrease in $\tau_i$ suggests that the $dq_i/dt$ caused by the deposition/sublimation process is accelerated (Morrison and Gettelman, 2008). Compared to the $1/\tau_i$ with $\mu_i = 0$ (i.e., $3.35\times10^{-4}$ s$^{-1}$ and $10.04\times10^{-4}$ s$^{-1}$), the $1/\tau_i$ with $\mu_i = 2$ ($4.66\times10^{-4}$ s$^{-1}$ and $13.98\times10^{-4}$ s$^{-1}$) and $\mu_i = 5$ ($5.25\times10^{-4}$ s$^{-1}$ and $15.74\times10^{-4}$ s$^{-1}$) increase by 39.10% and 56.72%, respectively. In Table 3, $N_{iauto}/N_i$ and $q_{iauto}/q_i$ indicate the portion of ICs that convert to snow in terms of number and mass, respectively. Under the small IC scenario (i.e., $R_{qi}$ = 20 µm), regardless of the value of $\mu_i$, both $N_{iauto}/N_i$ and $q_{iauto}/q_i$ are very small (< 2%, Table 3)





because there are few ICs with radii greater than $R_{cs}$ (Fig. 1). Under the large IC scenario (i.e., $R_{qi} = 60$ μm), there is a considerable portion of ICs with radii greater than $R_{cs}$, especially the mass contribution (Fig. 1). The $q_{iauto}/q_i$ at $\mu_i = 0$, 2, and 5 are 64.08%, 36.54%, and 18.40%, respectively (Table 3). This suggests that the autoconversion of IC to snow becomes

difficult with increasing $\mu_i$. Compared with the considerable values for $q_{iauto}/q_i$, the $N_{iauto}/N_i$ is relatively small (i.e., 4.84% at $\mu_i = 0$, 4.23% at $\mu_i = 2$, and 2.63% at $\mu_i = 5$). Therefore, the $R_{qi}$ of the residual ICs ($R_{qi\_afterauto}$; 43.36 μm at $\mu_i = 0$, 52.31 μm at $\mu_i = 2$, and 56.57 μm at $\mu_i = 5$) is obviously lower than the original $R_{qi}$ (60 μm). During the falling process, it is inevitable that $V_{qi}$ is greater than $V_{ni}$ because larger ICs with faster falling contribute more in the $V_{qi}$. Thus, larger ICs appear preferentially in the lower model layers. This is called the size-sorting mechanism (Milbrandt and Yau, 2005). $V_{qi}$ decreases with

increasing $\mu_i$, while $V_{ni}$ increases with increasing $\mu_i$ (Table 3). This could also be explained by their calculation equations (the corresponding derivations are similar to those for $R_{ni}$, $R_{ai}$, and $R_{ei}$, not shown). With increasing $\mu_i$, the difference between $V_{qi}$ and $V_{ni}$ decreases (Table 3) because the PSD of ICs becomes narrow (Fig.1). As a result, the size-sorting process becomes slow. For instance, there are many ICs with $R_{qi} = 60$ μm in a model layer. The height of each model layer is 200 m. After one model time step (10 min), some ICs fall into the lower layer. For $\mu_i = 0$, the $R_{qi}$ of the ICs that are still in the model layer

($R_{qi\_leftover}$) is 42.11 μm, and the $R_{qi}$ of the ICs in the lower layer ($R_{qi\_lowlayer}$) is 95.24 μm. For $\mu_i = 2$, $R_{qi\_leftover}$ is 52.45 μm, and $R_{qi\_lowerlayer}$ is 75.60 μm. For $\mu_i = 5$, $R_{qi\_leftover}$ is 55.81 μm, and $R_{qi\_lowerlayer}$ is 68.68 μm. It is clear that the difference in $R_{qi}$ between these two adjacent layers that is caused by the sedimentation process (i.e., the difference between $R_{qi\_leftover}$ and $R_{qi\_lowerlayer}$) becomes small with increasing $\mu_i$. In short, the above analyses clearly suggest that $\mu_i$ has a significant impact on the cloud microphysical processes and statistical mean radii of ICs.





**Table 3.** Off-line tests* for the cloud microphysical processes and statistical mean radii at $R_{qi} = 20\,\mu m$ (left) and $R_{qi} = 60\,\mu m$ (right).

| | $R_{qi} = 20\,\mu m$ | | | $R_{qi} = 60\,\mu m$ | | |
|---|---|---|---|---|---|---|
| | $\mu_i = 0$ | $\mu_i = 2$ | $\mu_i = 5$ | $\mu_i = 0$ | $\mu_i = 2$ | $\mu_i = 5$ |
| $R_{ni}$ (µm) | 11.00 | 15.33 | 17.26 | 33.02 | 45.98 | 51.78 |
| $R_{ai}$ (µm) | 15.57 | 17.70 | 18.64 | 46.70 | 53.09 | 55.93 |
| $R_{ei}$ (µm) | 33.02 | 25.54 | 23.01 | 99.06 | 76.63 | 69.04 |
| $1/\tau_i$ ($10^{-4}$ s$^{-1}$) | 3.35 | 4.66 | 5.25 | 10.04 | 13.98 | 15.74 |
| $\tau_i$ (s) | 2989.21 | 2146.68 | 1906.05 | 996.40 | 715.56 | 635.35 |
| $q_{iauto}/q_i$ (%) | 2.00 | 0.01 | 0 | 64.08 | 36.54 | 18.40 |
| $N_{iauto}/N_i$ (%) | 0.01 | 0 | 0 | 4.84 | 4.23 | 2.63 |
| $R_{qi\_afterauto}$ (µm) | 19.87 | 20.00 | 20.00 | 43.36 | 52.31 | 56.57 |
| $V_{qi}$ (cm s$^{-1}$) | 7.96 | 5.54 | 4.68 | 23.87 | 16.62 | 14.04 |
| $V_{ni}$ (cm s$^{-1}$) | 1.99 | 2.77 | 3.12 | 5.97 | 8.31 | 9.36 |

*Note $\tau_i$ is calculated at $T = 220$ K, $P = 330$ hPa, and $N_i = 10^5$ kg$^{-1}$ (~50 L$^{-1}$). $V_{ni}$ and $V_{qi}$ are calculated at $T = 220$ K and $P = 330$ hPa.

## 3.2 CAM6 simulations

During the evolution of stratiform clouds, the properties of ice clouds (e.g., $q_i$, $N_i$, and $R_{ni}$, including mixed-phase clouds) largely determine the ice-phase cloud microphysical processes. Meanwhile, these cloud microphysical processes in turn change the cloud properties. They interact as both cause and effect and finally reach climate equilibrium states. To facilitate the subsequent analyses, the cloud properties and $\mu_i$-related cloud microphysical processes are shown together in one figure. For ease of expression, "δ" is used to denote the changes in cloud properties that are caused by the cloud microphysical process during one model time step. For example, the changes in $q_i$ and $N_i$ that are caused by the sedimentation process during one model time step are denoted as $\delta q_{ised}$ and $\delta N_{ised}$, respectively.



**Figure 2.** Annual zonal mean in-cloud variables from the Mu0 (first and fourth columns), ΔMu2 (second and fifth columns) and ΔMu5 (third and sixth columns) experiments. Shown are the ICs mass mixing ratio ($q_i$) and number density ($N_i$), mass-weighted and number-weighted radii ($R_{qi}$ and $R_{ni}$), effective radius ($R_{ei}$), reciprocal of the supersaturation relaxation time scale ($1/\tau_i$), ice supersaturation ($S_i$), change in $q_i$ caused by deposition/sublimation process ($\delta q_{idep}$), portion of ICs that are converted to snow in terms of mass and number ($q_{iauto}/q_i$ and $N_{iauto}/N_i$), newly formed IC number density from the nucleation process ($\delta N_{inuc}$), the updated mass-weighted radius ($R_{qi}^*$) used for calculating the sedimentation process, mass-weighted and number-weighted fall velocities ($V_{qi}$ and $V_{ni}$), and changes in $q_i$ and $N_i$ that are caused by the sedimentation process ($\delta q_{ised}$ and $\delta N_{ised}$). Except for $\delta q_{idep}$, $\delta q_{ised}$, $\delta N_{inuc}$ and $\delta N_{ised}$, the other variables are shown as their relative changes (i.e., ΔMu2/Mu0 and ΔMu5/Mu0). The Y-axis indicates the atmospheric pressure (unit: hPa). The two black lines are the 0 and −37 ℃ isotherms. All results are sampled from model grids where the ice cloud fraction is greater than 1%. The shadows indicate that the differences between two experiments are not significant at the 95% level based on the Student's $t$ test.

Fig. 2 shows the model results from the Mu0, Mu2, and Mu5 experiments. The $q_i^{Mu0}$ is larger in the upper tropical troposphere ($> 3\ \mu g\ L^{-1}$) and relatively larger in the lower troposphere over middle latitudes in both hemispheres ($> 1\ \mu g\ L^{-1}$). The spatial pattern of $q_i^{Mu0}$ is generally in agreement with the satellite retrieval data (Li et al., 2012). Higher $N_i^{Mu0}$ ($> 200\ L^{-1}$) can be found in the tropopause region, where homogeneous freezing produces a large number of ICs (not shown) due to sufficient soluble aerosol particles, higher subgrid vertical velocity and lower temperature (Shi et al., 2015). All statistical





mean radii (i.e., $R_{qi}$, $R_{ni}$, and $R_{ei}$) decrease with altitude increasing. One possible reason is that it is hard for ICs to grow big in the upper troposphere because the water vapor density is very small over there (lower temperature). Furthermore, the size-

215 sorting effect (i.e., sedimentation process) could also be a contributor to this phenomenon (Milbrandt and Yau, 2005; Khain et al., 2015). As expected, $R_{ni}$ is less than $R_{qi}$, and $R_{ei}$ is larger than $R_{qi}$. After considering the impact of $\mu_i$ (i.e., $\mu_i = 2$ or 5), the $\Delta$Mu2 and $\Delta$Mu5 experiments show that $q_i$ is significantly increased while $N_i$ is significantly decreased. The $q_i^{\Delta Mu2/Mu0}$ is 30-100% in nearly all regions, and the $q_i^{\Delta Mu5/Mu0}$ reaches even higher levels (> 100%) in most regions. Both $N_i^{\Delta Mu2/Mu0}$ and $N_i^{\Delta Mu5/Mu0}$ are $< -20\%$ above the $-37°C$ isotherm and even reach $-50\%$ in the upper tropical troposphere. Consistent with the

220 increase in $q_i$ and the decrease in $N_i$, the $R_{qi}$ significantly increases. The $R_{qi}^{\Delta Mu2/Mu0}$ is 30-100% above the $-37°C$ isotherm, and the $R_{qi}^{\Delta Mu5/Mu0}$ is 30-100% in most regions and even reaches 100% in a few regions of the upper tropical troposphere. Because $R_{ni}$ increases with increasing $\mu_i$ at a fixed $R_{qi}$ value (Section 3.1), the relative increases in $R_{ni}$ from the $\Delta$Mu2 and $\Delta$Mu5 experiments (i.e., $R_{ni}^{\Delta Mu2/Mu0}$ and $R_{ni}^{\Delta Mu5/Mu0}$) are obviously higher than the relative increases in $R_{qi}$ (i.e., $R_{qi}^{\Delta Mu2/Mu0}$ and $R_{qi}^{\Delta Mu5/Mu0}$). The $R_{ni}^{\Delta Mu2/Mu0}$ is > 100% in some regions, and the $R_{ni}^{\Delta Mu5/Mu0}$ is > 100% in most regions. Compared with

225 the relative increases in $R_{qi}$, the relative increases in $R_{ei}$ from the $\Delta$Mu2 and $\Delta$Mu5 experiments are obviously reduced or even negative because $R_{ei}$ decreases with increasing $\mu_i$ at a fixed $R_{qi}$ value (Section 3.1). Overall, the impacts of $\mu_i$ on $q_i$ and $N_i$ are notable. The changes in the statistical mean radii (i.e., $R_{qi}$, $R_{ni}$, and $R_{ei}$) can be explained by the changes in $q_i$, $N_i$, and $\mu_i$.

This paragraph analyzes the interaction between the ice cloud properties ($q_i$, $N_i$, and $R_{ni}$) and the IC deposition/sublimation process, and the influence of $\mu_i$ on this interaction. Since $1/\tau_i$ is proportional to $N_i R_{ni}$ (Table 1), the $1/\tau_i^{Mu0}$ is larger in the

230 upper tropical troposphere (> $20\times10^{-4}$ s$^{-1}$) due to the high $N_i^{Mu0}$ (> 200 L$^{-1}$). Both the $\Delta$Mu2 and $\Delta$Mu5 experiments show that the $1/\tau_i$ increases in most regions because the relative increase in $R_{ni}$ (i.e., $R_{ni}^{\Delta Mu2/Mu0}$ and $R_{ni}^{\Delta Mu5/Mu0}$) is stronger than the relative decrease in $N_i$ (i.e., $N_i^{\Delta Mu2/Mu0}$ and $N_i^{\Delta Mu5/Mu0}$). However, the $1/\tau_i$ is slightly decreased in some regions of the upper tropical troposphere because the relative decrease in $N_i$ is remarkable (< $-50\%$) in these regions. The $\delta q_{idep}$ which indicates the change in $q_i$ caused by the deposition/sublimation process is mainly determined by the $1/\tau_i$ and in-cloud ice

supersaturation ($S_i$) (Morrison and Gettelman, 2008). Except for a very small region, the annual zonal mean $S_i^{Mu0}$ is positive. This is consistent with the deposition events being much more frequent than sublimation events (not shown). When $S_i > 0$, ice-supersaturation (i.e., $S_i > 0$) towards ice-saturation (i.e., $S_i = 0$) occurs because the water vapor is consumed by $\delta q_{idep}$ (Korolev et al., 2003; Krämer et al., 2009). The $S_i^{Mu0}$ is lower (< 3%) in the upper tropical troposphere due to the high $1/\tau_i^{Mu0}$ (> $20\times10^{-4}$ s$^{-1}$). Both the $\Delta$Mu2 and $\Delta$Mu5 experiments show that $S_i$ is increased in the upper tropical troposphere due to the

decreasing $1/\tau_i$, and $S_i$ is decreased in the other regions due to the increasing $1/\tau_i$. It is noteworthy that the $S_i^{\Delta Mu2}$ and $S_i^{\Delta Mu5}$ in the mixed-phase cloud layers are obviously weaker than those in the pure ice cloud layers (i.e., above the $-37°C$ isotherm). This is consistent with that the $S_i$ is relatively stable in mixed-phase clouds because liquid droplets are often present. The $\delta q_{idep}^{Mu0}$ is generally decreased with the altitude because the saturated vapor pressure significantly decreases with decreasing air temperature. The comparison between $\delta q_{idep}$ and $q_i$ suggests that $\delta q_{idep}$ is an important source of $q_i$. The $\delta q_{idep}^{\Delta Mu2}$ and

$\delta q_{idep}^{\Delta Mu5}$ are greater than 0.1 µg L$^{-1}$ in most mixed-phase cloud layers due to the strongly increasing $1/\tau_i$ and relatively stable





$S_i$ values. This suggests that the increasing $\mu_i$ could lead to a higher equilibrium state of $q_i$ in the mixed-phase cloud layers via the deposition process. The $\delta q_{idep}{}^{\Delta Mu2}$ and $\delta q_{idep}{}^{\Delta Mu5}$ are negative between 200 hPa and 300 hPa mainly because the $S_i{}^{\Delta Mu2}$ and $S_i{}^{\Delta Mu5}$ are negative and the $1/\tau_i{}^{\Delta Mu2/Mu0}$ and $1/\tau_i{}^{\Delta Mu5/Mu0}$ are relatively small. The $\delta q_{idep}{}^{\Delta Mu2}$ and $\delta q_{idep}{}^{\Delta Mu5}$ are positive above 100 hPa mainly because the $S_i{}^{\Delta Mu2}$ and $S_i{}^{\Delta Mu5}$ are positive. These results indicate that the impact of $\mu_i$ on $\delta q_{idep}$ becomes

complex above the −37°C isotherm, where $S_i$ is more susceptible to $1/\tau_i$ and $\delta q_{idep}$. Meanwhile, the impact of $\mu_i$ on $\delta q_{idep}$ also becomes weak above the −37°C isotherm because the feedback processes (i.e., the interaction between $S_i$ and $\delta q_{idep}$) become important. In short, the $\mu_i$-induced changes in the deposition/sublimation process (i.e., $1/\tau_i$ and $\delta q_{idep}$) can be largely explained by the changes in $N_i$ and $R_{ni}$. One reason for the higher $q_i$ in the mixed-phase cloud layers from the Mu2 and Mu5 experiments is that $\delta q_{idep}$ increases with increasing $\mu_i$.

This paragraph analyzes the interaction between the ice cloud properties ($q_i$, $N_i$, and $R_{qi}$) and the autoconversion of IC to snow process (hereafter the autoconversion process), and the influence of $\mu_i$ on this interaction. Both $q_{iauto}/q_i{}^{Mu0}$ and $N_{iauto}/N_i{}^{Mu0}$ are decreased with the altitude because the $R_{qi}{}^{Mu0}$ is decreased with the altitude. As expected, the $q_{iauto}/q_i{}^{Mu0}$ is considerable and much larger than the $N_{iauto}/N_i{}^{Mu0}$. It is clear that the autoconversion process is an important sink of $q_i$. However, the autoconversion process is not an important sink of $N_i$ because the $N_{iauto}/N_i$ is very small. Both the ΔMu2 and

ΔMu5 experiments show that the $q_{iauto}/q_i$ is significantly decreased because the autoconversion process obviously becomes difficult at higher $\mu_i$ values (off-line tests, Section 3.1). The difficult autoconversion process leads to an equilibrium state with higher $q_i$ and larger $R_{qi}$. Because of the larger $R_{qi}$, the $N_{iauto}/N_i$ from the Mu2 and Mu5 experiments are significantly increased. The increasing $N_{iauto}/N_i$ from the Mu2 and Mu5 experiments might be a main reason for the decrease in $N_i$ in the mixed-phase cloud layers. However, the remarkable decrease in $N_i$ (mostly in the pure ice cloud layers) from the Mu2 and

Mu5 experiments is mainly due to the ice nucleation process. In the MG scheme, the newly formed IC number density (excluding the ICs in mixed-phase clouds) is calculated by a physically based ice nucleation parameterization (Liu and Penner, 2005). Because the autoconversion process becomes difficult in the Mu2 and Mu5 experiments, the in-cloud ICs should have longer lifetimes and larger radii. As a result, $\delta N_{inuc}$, which denotes the newly formed IC number density from the nucleation process, significantly decreases in the Mu2 and Mu5 experiments (Fig. 2). The main reason is that the pre-

existing ICs would hinder the subsequent ice nucleation process (especially for homogeneous freezing) owing to the depletion of water vapor via deposition growth (Barahona et al., 2014; Shi et al., 2015). $\delta N_{inuc}$ is the main source of $N_i$. Therefore, both the ΔMu2 and ΔMu5 experiments show that $N_i$ is significantly decreased. In short, the increase in $\mu_i$ causes the autoconversion process to be difficult and then leads to a higher equilibrium state of $q_i$ and $R_{qi}$. Meanwhile, $N_i$ is significantly decreased due to the higher equilibrium state of $q_i$ and $R_{qi}$ (i.e., the stronger suppression effect of the pre-

existing ICs on the ice nucleation process).

This paragraph analyzes the interaction between the ice cloud properties and the IC sedimentation process, and the influence of $\mu_i$ on this interaction. The sedimentation process is the last cloud microphysical process in the MG scheme. The IC fall velocity is calculated based on the updated cloud properties (i.e., the other cloud microphysical processes at this model time





step have been considered). Here, $R_{qi}^*$ denotes the updated $R_{qi}$. In the mixed-phase cloud layers, the $R_{qi}^{*Mu0}$ is slightly less

than the $R_{qi}^{Mu0}$ because the sedimentation process has not occurred. After considering the impacts of $\mu_i$ on the cloud microphysical processes introduced above, the relative increases in $R_{qi}^*$ from the ΔMu2 and ΔMu5 experiments (i.e., $R_{qi}^{*\Delta Mu2/Mu0}$ and $R_{qi}^{*\Delta Mu5/Mu0}$) are higher than the relative increase in $R_{qi}$. As expected, both $V_{qi}^{Mu0}$ and $V_{ni}^{Mu0}$ decrease with altitude increasing, and the $V_{qi}^{Mu0}$ is obviously larger than the $V_{ni}^{Mu0}$. Although the $R_{qi}^{*\Delta Mu2/Mu0}$ and $R_{qi}^{*\Delta Mu5/Mu0}$ are positive, the $V_{qi}^{\Delta Mu2/Mu0}$ and $V_{qi}^{\Delta Mu5/Mu0}$ are negative in most regions because $V_{qi}$ decreases with increasing $\mu_i$ (off-line tests, Section

3.1). However, both $V_{qi}^{\Delta Mu2/Mu0}$ and $V_{qi}^{\Delta Mu5/Mu0}$ are positive in some layers over the tropics and subtropics, where the $R_{qi}^{*\Delta Mu2/Mu0}$ and $R_{qi}^{*\Delta Mu5/Mu0}$ are relatively higher. Because $V_{ni}$ increases with increasing $\mu_i$ at a fixed $R_{qi}$ value (off-line tests, Table 3) and the $R_{qi}^*$ from the Mu2 and Mu5 experiments are increased, the relative increases in $V_{ni}$ from the ΔMu2 and ΔMu5 experiments are remarkable. The $V_{ni}^{\Delta Mu2/Mu0}$ is > 100% in some regions, and the $V_{ni}^{\Delta Mu5/Mu0}$ is > 100% in most regions. $\delta q_{ised}$ is mainly determined by the gradient of $V_{qi}q_i$ in the vertical direction. Actually, the newly updated $q_i$ between the

substeps of the sedimentation process is used for calculating $\delta q_{ised}$. Similarly, $\delta N_{ised}$ is mainly determined by the gradient of $V_{ni}N_i$ in the vertical direction. Furthermore, the ICs that fall into the clear portions of the lower model layer sublimate instantly. Therefore, both $\delta q_{ised}^{Mu0}$ and $\delta N_{ised}^{Mu0}$ are negative in most regions. This is consistent with sedimentation being a sink of clouds. The $\delta q_{ised}$ from the Mu2 and Mu5 experiments (i.e., $\delta q_{ised}^{\Delta Mu2}$ and $\delta q_{ised}^{\Delta Mu5}$) decrease (negative, stronger sink) in most regions mainly because of the increasing $V_{qi}$ and higher $q_i$. The $\delta N_{ised}$ from the Mu2 and Mu5 experiments (i.e.,

$\delta N_{ised}^{\Delta Mu2}$ and $\delta N_{ised}^{\Delta Mu5}$) increase (positive, weaker sink) in a few layers over the tropics. This is mainly due to the changes in the vertical gradient of $N_i$. Both the ΔMu2 and ΔMu5 experiments show that the changes in $\delta q_{ised}$ and $\delta N_{ised}$ are generally weaker than the changes in $\delta q_{idep}$, $\delta q_{iatuo}$ (i.e., $q_i \times q_{iauto}/q_i$), and $\delta N_{inuc}$. In short, the fall velocities (i.e., $V_{qi}$ and $V_{ni}$) and their impacts on ice clouds (i.e., $\delta q_{ised}$ and $\delta N_{ised}$) are mainly determined by the cloud properties (i.e., $q_i$, $N_i$, $R_{qi}$, and $R_{qi}^*$). Although the sedimentation process is also a main factor that determines the cloud properties, the changes in the

sedimentation process that are caused by the increasing $\mu_i$ are not as strong as those in the deposition/sublimation, autoconversion, and nucleation processes.

Based on the analyses presented above, it can be concluded that increasing $\mu_i$ would lead to a climate equilibrium state with higher $q_i$ and lower $N_i$ in most regions. The changes in the statistical mean radii (i.e., $R_{qi}$, $R_{ni}$, and $R_{ei}$) and ice-phase cloud microphysical processes (i.e., $\delta q_{idep}$, $q_{iauto}/q_i$, $N_{iauto}/N_i$, $\delta N_{inuc}$, $\delta q_{ised}$ and $\delta N_{ised}$) are mainly determined by the higher $q_i$, lower

$N_i$, and increasing $\mu_i$. On the other hand, the higher $q_i$ and lower $N_i$ can largely be explained by the changes in the ice-phase cloud microphysical processes (i.e., $\delta q_{idep}$, $q_{iauto}/q_i$, and $\delta N_{inuc}$) that are caused by the increasing $\mu_i$. Furthermore, the ΔMu2 and ΔMu5 experiments show very similar spatial patterns for the $\mu_i$-induced changes. This suggests that the impact of $\mu_i$ on the simulated climate equilibrium state is stable.



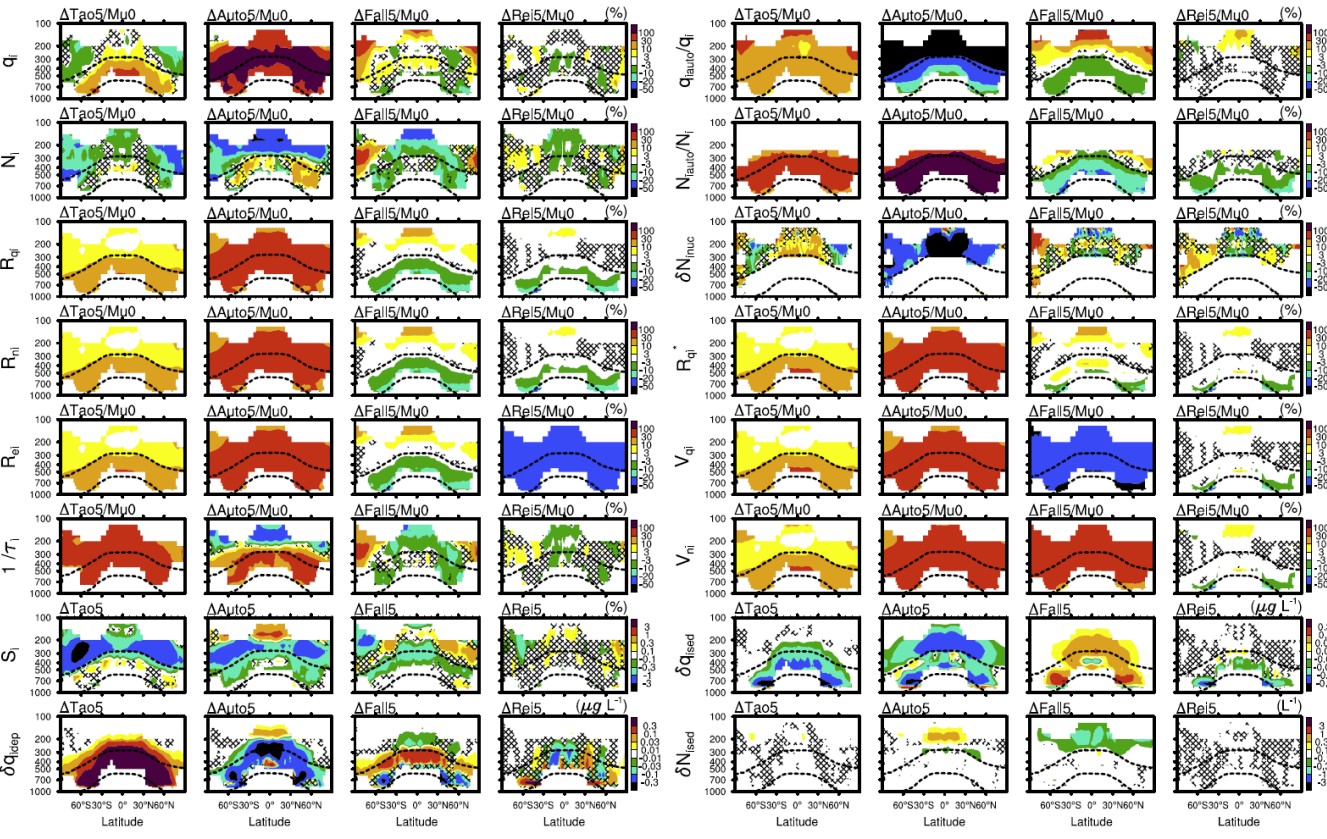

**Figure 3.** Similar to Fig. 2 but for the ΔTao5, ΔAuto5, ΔFall5 and ΔRei5 experiments. Except for the $R_{ei}^{Rei5}$, all statistical mean radii are calculated with $\mu_i = 0$.

Fig. 3 shows the changes in the simulated climate equilibrium states that are caused by each individual $\mu_i$-related process. The $1/\tau_i$ from the Tao5 experiment is significantly increased ($1/\tau_i^{\Delta Tao5/Mu0}$). Similar to the ΔMu2 and ΔMu5 experiments, this

increasing $1/\tau_i$ could lead to a higher equilibrium state of $q_i$ in the mixed-phase cloud layers ($q_i^{\Delta Tao5/Mu0}$) via the deposition/sublimation process ($\delta q_{idep}^{\Delta Tao5}$). However, this increasing $1/\tau_i$ leads to lower $q_i$ and lower $N_i$ in most of the pure ice cloud layers. The main reason might be that the ICs grow faster and their lifetimes become shorter. The $q_{iauto}/q_i$ from the Auto5 experiment is significantly decreased ($q_{iauto}/q_i^{\Delta Tao5/Mu0}$). This could lead to a higher $q_i$ in nearly all regions ($q_i^{\Delta Auto5/Mu0}$) and a lower $N_i$ in the pure ice cloud layers ($N_i^{\Delta Auto5/Mu0}$). The mechanism is the same as that introduced based on the ΔMu2

and ΔMu5 experiments. It is noteworthy that the $N_i$ from the Auto5 experiment is slightly increased in some mixed-phase cloud layers. The main reason might be that the accretion of $N_i$ by snow is significantly decreased in the mixed-phase cloud layers (not shown) due to the difficult autoconversion process. The $V_{qi}$ from the Fall5 experiment is significantly decreased ($V_{qi}^{\Delta Fall5/Mu0}$), and the sink term of $q_i$ due to sedimentation becomes weaker (i.e., positive $\delta q_{ised}^{\Delta Fall5}$) in most regions. Unlike the $V_{qi}^{\Delta Fall5/Mu0}$, the $V_{ni}$ from the Fall5 experiment obviously increase ($V_{ni}^{\Delta Fall5/Mu0}$), and the sink term of $N_i$ due to

sedimentation becomes stronger (i.e., negative $\delta N_{ised}^{\Delta Fall5}$) in the pure ice cloud layers. These might be the main reasons for



the increase in $q_i$ and the decrease in $N_i$ in the pure ice cloud layers over the tropics ($q_i^{\Delta Fall5/Mu0}$ and $N_i^{\Delta Fall5/Mu0}$). It is interesting to note that the $R_{qi}$, $R_{ni}$, and $R_{ei}$ from the Fall5 experiment all increase in pure ice cloud layers (i.e., upper layers) and decrease in mixed-phase cloud layers (i.e., lower layers). This can be explained by the $\mu_i$-induced weaker size-sorting mechanism (Section 3.1). The $R_{ei}$ from the Rei5 experiment ($R_{ei}^{\Delta Rei5/Mu0}$) is significantly decreased. Because the change of $R_{ei}$

does not directly affect the cloud microphysical processes, the changes in cloud properties from the ΔRei5 experiment are not statistically significant in most regions. Taken overall, the above analyses clarify the mechanism of $\mu_i$'s impacts. Increasing $\mu_i$ in autoconversion impacts pure ice clouds the most (i.e., significantly increased $q_i$ and significantly decreased $N_i$ in the pure ice cloud layers). Furthermore, increasing $\mu_i$ in autoconversion also leads to a much higher $q_i$ in the mixed-phase cloud layers. Increasing $\mu_i$ in deposition/sublimation can also lead to a higher $q_i$ in the mixed-phase cloud layers.

Increasing $\mu_i$ in sedimentation can lead to a higher IC radius in the upper layers and lower IC radius in the lower layers. The impacts from sedimentation and deposition/sublimation are obviously weaker than those from autoconversion. The changes caused by increasing $\mu_i$ in the radiative process (i.e., $R_{ei}$) are relatively chaotic.

The above analyses focus on cloud properties and cloud microphysical processes (i.e., in-cloud variables). This paragraph discusses the impacts of $\mu_i$ on radiation and precipitation. The annual zonal mean distributions of the ice water path (IWP),

column $N_i$ (ColN$_i$), longwave (CRE$_{LW}$) and shortwave (CRE$_{SW}$) cloud radiative effects, and convective (RainC) and large-scale (RainL) precipitation rates are shown in Fig. 4, and the corresponding global annual mean values are listed in Table 4. The comparison of the Mu0, Mu2 and Mu5 experiments shows that the zonal mean IWPs over all latitudes clearly increase with increasing $\mu_i$. This is consistent with the changes in in-cloud $q_i$ (Fig. 2). The comparison of the ΔMu5, ΔTao5, ΔAuto5, ΔFall5 and ΔRei5 experiments shows that the $\mu_i$-induced increases in IWP are mainly provided by the autoconversion

process. Compared to the Mu0 experiment, the ColN$_i$ from the Mu2 and Mu5 experiments obviously decrease over tropical regions. It is clear that the autoconversion process is also the main contributor to the decreases in ColN$_i$ (Fig. 4, right column). Compared to the Mu0 experiment, both CRE$_{LW}$ and CRE$_{SW}$ from the Mu2 and Mu5 experiments are obviously enhanced mainly because of the increasing IWPs. It is clear that the enhancements of CRE$_{LW}$ and CRE$_{SW}$ are also mainly contributed to by the autoconversion process (Fig. 4). Both the CRE$_{LW}$ and CRE$_{SW}$ from the Rei5 experiment are also

obviously enhanced in terms of their zonal mean values (Fig. 4) and global mean values (Table 4, CRE$_{LW}^{\Delta Rei5}$ = 1.29 W m$^{-2}$ and CRE$_{SW}^{\Delta Rei5}$ = −1.79 W m$^{-2}$). This suggests that the impact of $\mu_i$ on $R_{ei}$ could lead to considerable changes in the Earth's radiation budget. Compared to the impacts of $\mu_i$ on radiation, the impact on large-scale precipitation (i.e., RainL) is not statistically significant (Fig. 4, left column). However, the convective precipitation from the ΔMu5 experiment (i.e., RainC$^{\Delta Mu5}$) is significantly reduced over the tropics and subtropics (Fig. 4, right column). The reason is that the increase in

ice clouds (i.e., $q_i$) increases atmospheric stability via the radiative budget and then leads to weaker convective precipitation (Andrews et al., 2010; Wang et al., 2014). Overall, the impacts of $\mu_i$ on radiation and precipitation are considerable. The global mean CRE$_{LW}^{\Delta Mu5}$, CRE$_{SW}^{\Delta Mu5}$ and RainC$^{\Delta Mu5}$ are 5.58 W m$^{-2}$, −5.34 W m$^{-2}$ and −0.12 mm day$^{-1}$, respectively. These changes are mainly contributed to by the autoconversion process. Furthermore, the comparisons between the ΔMu2 and

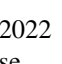


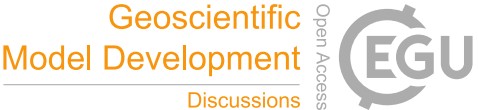

ΔMu5 experiments (Fig. 4 and Table 4) show that, in most cases, the $\mu_i$-induced changes are enhanced with increasing $\Delta\mu_i$.
This suggests that, in terms of the zonal mean and global mean values, the impacts of $\mu_i$ are relatively stable.

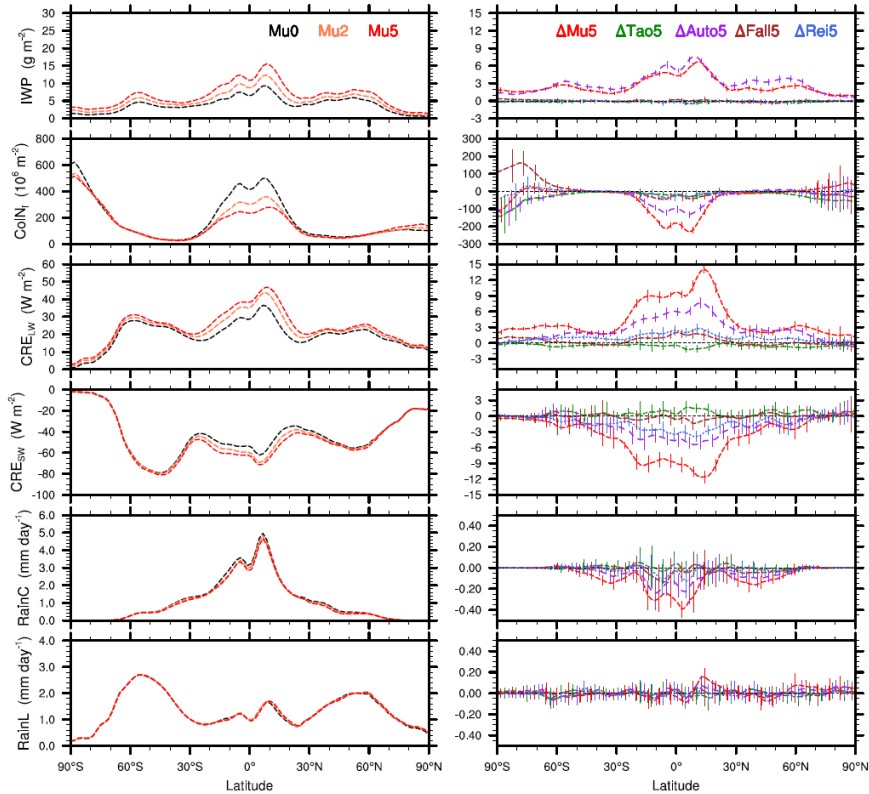

**Figure 4.** Annual zonal mean distributions of the ice water path (IWP), column $N_i$ (ColN$_i$), longwave and shortwave cloud radiative effects (CRE$_{LW}$ and CRE$_{SW}$), and convective and large-scale precipitation rates (RainC and RainL). In the right panel (i.e., Δ), the vertical bars overlying these lines indicate the ranges of two standard deviations from the difference of each of 10 years at different latitudes.

**Table 4.** The global annual mean variables are shown in Fig. 4. The corresponding standard deviations calculated from the difference of each year for 10 years are shown in brackets.

| Experiments | Mu0 | ΔMu2 | ΔMu5 | ΔTao5 | ΔAuto5 | ΔFall5 | ΔRei5 |
|---|---|---|---|---|---|---|---|
| IWP (g m$^{-2}$) | 4.74 | 1.37(0.10) | 2.97(0.06) | −0.20(0.05) | 3.64(0.06) | −0.01(0.04) | −0.09(0.06) |
| ColN$_i$ ($10^8$ m$^{-2}$) | 1.89 | −0.41(0.06) | −0.56(0.04) | −0.15(0.03) | −0.37(0.03) | −0.06(0.04) | −0.07(0.05) |
| CRE$_{LW}$ (W m$^{-2}$) | 22.22 | 3.40(0.12) | 5.58(0.13) | −0.52(0.07) | 3.40(0.11) | 0.49(0.09) | 1.29(0.10) |
| CRE$_{SW}$ (W m$^{-2}$) | −49.25 | −3.00(0.43) | −5.34(0.31) | 0.54(0.38) | −2.52(0.46) | −0.20(0.36) | −1.79(0.24) |
| RainC (mm day$^{-1}$) | 1.57 | −0.08(0.01) | −0.12(0.01) | 0 (0.01) | −0.07(0.01) | −0.02(0.01) | −0.02(0.01) |
| RainL (mm day$^{-1}$) | 1.39 | 0.01(0.01) | 0.01(0.01) | 0 (0.01) | 0.01(0.01) | 0 (0) | 0 (0.01) |



## 4 Summary and conclusions

This paper investigates the impacts of $\mu_i$ on climate simulations with the CAM6 model. To achieve this, the two-moment bulk cloud microphysics scheme used in CAM6 was modified to consider the $\mu_i$'s impacts by a tunable parameter. After that, the impacts of $\mu_i$ on the IC size distribution and $\mu_i$-related cloud microphysical processes are illustrated first by calculation equations and off-line tests, and the impacts of $\mu_i$ on the climate simulations are then analyzed with the CAM6 model.

The impacts of $\mu_i$ on the IC size distribution and various statistical mean radii are clearly explained by the calculation equations. $R_{qi}$ can be calculated from $q_i$ and $N_i$, and the normalized IC size distribution (i.e., the relative contributions from each bin) can be calculated from $R_{qi}$ and $\mu_i$. The impacts of $\mu_i$ on mass-weighted size distribution is obviously different from that on the commonly used number-weighted size distribution (Fig.1). Unlike the number contributions, the mass contributions of the larger size bins significantly decrease with increasing $\mu_i$ because the mass contribution is more sensitive to the IC radius. In the bulk cloud microphysics scheme, the physical processes are calculated based on various statistical mean radii. The ratios of the other statistical mean radii (i.e., $R_{ni}$, $R_{ai}$, and $R_{ei}$) to $R_{qi}$ are functions of $\mu_i$. At $\mu_i \geq 0$, $R_{ni}$ and $R_{ai}$ are always less than $R_{qi}$, whereas $R_{ei}$ is always greater than $R_{qi}$. The differences among these statistical mean radii become small with increasing $\mu_i$, which is consistent with the narrower size distribution determined by higher $\mu_i$ values.

The impacts of $\mu_i$ on the $\mu_i$-related cloud microphysical processes are clearly illustrated by the off-line tests. Under the same $q_i$ and $N_i$ ($R_{qi}$ is also fixed), the IC deposition/sublimation process is considerably accelerated with increasing $\mu_i$ because the $R_{ni}$ used for calculating deposition/sublimation is increasing with increasing $\mu_i$. Under the same $R_{qi}$ ($R_{qi}$ is much less than the snow radius), the autoconversion of IC to snow obviously becomes difficult with increasing $\mu_i$ because the portion of ICs with radii greater than the threshold ($> R_{cs}$) decreases under the narrow size distribution. A major effect of IC sedimentation is size-sorting because $V_{qi}$ is greater than $V_{ni}$, and $\mu_i$ plays an important role in determining the rate of size-sorting (Milbrandt and Yau, 2005). In this study, the off-line tests clearly show that the difference in IC radius between two adjacent model layers caused by sedimentation becomes small with increasing $\mu_i$ because the difference between $V_{qi}$ and $V_{ni}$ becomes small (i.e., the size-sorting rate becomes slow).

The climate simulations show that the impacts of $\mu_i$ on the ice cloud properties are notable, and the main corresponding mechanisms are clear. After increasing $\mu_i$ from 0 to 2 and 5, $q_i$ significantly increases, while $N_i$ significantly decreases. The accelerated deposition process contributes to the higher $q_i$ in the mixed-phase cloud layers where the ice supersaturation ($S_i > 0$) is relatively stable. The difficult autoconversion process leads to longer IC lifetime and higher $q_i$. Meanwhile, $N_i$ significantly decreases because the newly formed IC number density is significantly decreased owing to the longer IC lifetime and higher $q_i$. The experiments with only one modified $\mu_i$-related process make the mechanisms of $\mu_i$'s impacts more clear. Autoconversion contributes the most. The $\mu_i$ also has considerable impacts on radiation and precipitation. After increasing $\mu_i$ from 0 to 5, the global mean $CRE_{LW}$ is increased (stronger warming effect) by 5.58 W m$^{-2}$ (25.11%). Meanwhile, the $CRE_{SW}$ is decreased (less cooling effect) by $-5.34$ W m$^{-2}$ (10.84%). The enhancement of the cloud radiative

effects is largely provided by the higher equilibrium state of $q_i$. The considerably stronger $CRE_{LW}$ could increase the atmospheric stability and then lead to weaker convective precipitation (Andrews et al., 2010; Wang et al., 2014). As expected, after increasing $\mu_i$ from 0 to 5, the global mean RainC is decreased by $-0.12$ mm day$^{-1}$ (7.64%). In short, the impacts of $\mu_i$ on climate simulations are significant. This suggests that the $\mu_i$ (i.e., the PSD of ICs) and $\mu_i$-related cloud microphysical processes deserve a more realistic representation in climate models, especially for cloud schemes with autoconversion. Fortunately, there have been some studies that can help to address this issue. Such as, $\mu_i$ is described by an empirical formula (Eidhammer et al., 2017), $\mu_i$ is prognosted in a three-moment cloud scheme (Milbrandt et al., 2021), and single-ice-category cloud schemes could obviate the need for autoconversion process (e.g., Morrison and Milbrandt, 2015; Eidhammer et al., 2017; Zhao et al., 2017).

**Code and data availability**

The CAM6 model used in this study is the atmospheric component of the Community Earth System Model version 2.1.3 (CESM2.1.3). The CESM2.1.3 is a release version of CESM2. The model code, scripts, and input data are freely available through a public GitHub repository (https://escomp.github.io/CESM/versions/cesm2.1/html/downloading_cesm.html; last access: 9 May 2022). More details about model workflow can be found in the corresponding quick-start guide. The modified model code, model run control scripts, and postprocessing scripts covering every data processing action for all the model results reported in the paper are available online at DOI: 10.5281/zenodo.6530482. The FORTRAN code for offline tests is also archived with the same DOI. Furthermore, the NCL scripts and data for making every figure are also available at the above DOI.

**Competing interests**

The authors declare that they have no conflict of interest.

**Author contribution**

XS designed this study. WZ and XS designed the CAM6 model experiments and developed the modified model code. XS and WZ analyzed the results and wrote the original paper. All authors contributed to improving and reviewing the manuscript.

**Acknowledgments**

This study was supported by the National Key Research and Development Program of China (grant no. 2017YFA0604001) and the National Natural Science Foundation of China (grant no. 41775095). The authors would like to thank Yuxi Zeng and



Jiaojiao Liu for checking the English expressions of this manuscript. The model simulation was conducted in the High Performance Computing Center of Nanjing University of Information Science & Technology.

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
