# Peer review of "Impacts of Ice-Particle Size Distribution Shape Parameter on Climate Simulations with the Community Atmosphere Model Version 6 (CAM6)"

_Geoscientific Model Development, 2022_

## Referee Comment (RC1)

Review for Geoscientific Model Development (GMD)
5 July 2022

Title: Impacts of Ice-Particle Size Distribution Shape Parameter on Climate Simulations with the Community Atmosphere Model Version 6 (CAM6)
Author(s): Wentao Zhang et al.
MS No.: gmd-2022-95
MS type: Model evaluation paper

General Comments:

Global Climate Models (GCMs) simulate ice clouds using "two-moment schemes" having prognostic solutions for ice particle number concentration ($N_i$) and cloud ice water content (IWC). The ice particle size distribution (PSD) is generally parameterized as a gamma PSD as described in Eq. 1 in this manuscript, consisting of three variables: $N_0$, $\lambda$ and $\mu$. Knowing $N_i$ and IWC, $N_0$ and $\lambda$ are solved for, but in order to obtain mathematical closure for the PSD, $\mu$ is given an arbitrary constant value. To my knowledge, there are very few papers investigating the impact of $\mu$ on ice cloud microphysical processes in climate models, and this paper appears to be the most thorough and appropriate for GCMs to date. The paper is well organized and well written, providing important new findings relevant to a GMD readership.

In the study by Mitchell et al. (2006, Atmos. Res.), a snow growth model was developed and tested against aircraft PSD probe measurements where the aircraft descends from cloud top to cloud base under quasi-steady state widespread snowfall conditions, where PSDs were modeled as gamma PSDs. Optimal agreement between the height-evolution of measured and predicted PSDs was obtained for a $\mu$ value of -0.6. Assuming the snow growth model was developed properly, this suggests that the PSDs sampled on this flight were characterized by slightly negative $\mu$. Other studies (e.g., Herzegh and Hobbs, 1985, QJRMS; Gordon and Marwitz, 1986, JAS; Mitchell, 1988, JAS) suggest $\mu$ typically ranges between 1 and -1 in ice clouds, while Heymsfield (2003, JAS, Part 2) finds $\mu$ lies mostly between -2 and 2 when natural ice PSDs are parameterized as gamma PSDs. Thus, the $\mu$ values of 2 and 5 assumed for ice clouds in this study appear atypical, but the impacts of changing $\mu$ from 0 to 2 (shown in this study) are relevant to real cloud microphysical and radiative processes. This paper would be much more realistic and useful if it also evaluated the impact of changing $\mu$ from 0 to -1. Negative values of $\mu$ are common when ice crystal nucleation rates are relatively high (Herzegh and Hobbs, 1985, QJRMS).

Mitchell (1991, JAS) found that for negative $\mu$ values (i.e., superexponential PSD), aggregation was the only growth process that substantially increased ice particle sizes, whereas for positive $\mu$, both aggregation and vapor diffusion contributed to ice particle size increases. This is an example of how ice particle growth processes act differently depending on the sign of $\mu$ and illustrates the need to consider both positive and negative $\mu$ values.

Major Comments:

1. Line 68: Sentence states that $\mu_i$ is not considered in the default MG scheme because $\mu_i$ = 0. While it is true that $\mu_i$ = 0, stating that it was not considered is misleading. On a number of occasions, Hugh Morrison indicated to me that he was seeking more information about $\mu_i$ and was exploring new ways of treating it (prior to the release of CAM5). After consulting with his peers, he decided a value of zero was most reasonable if a fixed value was to be used.

2. Table 1: In the MG scheme, the air density prefactor for the mass-weighted ice fall speed is raised to the power of 0.54 (following Heymsfield and Bansemer 2007), not 0.35 as shown in Table 1. The simulations may need to be rerun if this incorrect value of 0.35 was used.

3. Line 317: There are references to support this statement; please add some.

4. Lines 381-383: This finding appears similar to that reported in Mitchell (1991, JAS) titled "Evolution of snow size spectra in cyclonic storms. Part II: Deviations from the exponential form", where it was found that the IC vapor deposition process was accelerated by increasing $\mu_i$.

Technical Comments:

1. Line 47: Common Atmosphere Model => Community Atmosphere Model?
2. Line 89: PDF => PSD?
3. Caption for Fig. 2 near bottom: "The two black lines" => "The two black dashed curves"?
4. Lines 279-280: Are the superscripts for *Mu and Mu correct?

---

## Referee Comment (RC2)

*Zhang et al.* investigate the impact of the shape parameter of cloud ice on simulated cloud properties and radiation using Community Atmosphere Model Version 6 (CAM6) of Community Earth System Model Version 2. The shape parameter is one of three parameters of the gamma distribution, which is commonly used in the two-moment bulk cloud microphysics scheme. Considering the current microphysics schemes usually set shape parameter to 0, investigate the impact of different values of shape parameter on the cloud properties providing important findings relevant to a GMD readership.

The authors conducted offline analysis and 10-year simulation using different value (0-5) of the shape parameter, in order to show how the shape parameter influences the cloud properties and radiation transfer. The authors suggested that increasing the value of the shape parameter would lead to higher qi and lower Ni in most regions globally, furthermore, the longwave cloud radiative forcing increases by 5.58 W m$^{-2}$ (25.11%), and the convective precipitation rate decreases by −0.12 mm day$^{-1}$ (7.64%).

After serious consideration, I here recommend this manuscript subject to major revisions

**My major concerns are as below:**

1. If I understand correctly, the authors only added the shape parameter to the PSD of cloud ice, the PSD of snow is not considered. This should be indicated in the manuscript, because "ice crystal" represents all the ice particles (cloud ice and snow togther).

2. Observation data of ice water path (IWP), shortwave cloud forcing (SWCF), and longwave cloud forcing (LWCF) are need in Figure 4, Table 4, and related analysis. This will give important information of which value of shape parameter give more realistic simulation results compared with observation.

3. From Figure 4, the IWP, LWCF, and SWCF have dramatic changes over the tropics, with much smaller changes in the mid-and high latitude. As we know most of the cloud and precipitation process over the tropics is governed by the convection scheme.

Since the shape parameter is only introduced in the large-scale microphysics, while the convection scheme is not changed.    Why do the IWP, precipitation, and cloud forcing have the largest change over the tropics? The reader may expect that the largest change is notiecd in the mid-and high latitude, where larger-scale microphysics treated most of the cloud process, therefore, your modifications should have a larger impact over there. Considering the convection precipitation changed also largely, this may indicate the change of IWP, precipitation, and cloud forcing is directly caused by the convection process (as a result of changed climate state), not by the shape parameter in the microphysics scheme.

**Minor comments:**

1. Line 22: "cloud-related process", to be consistent with the beginning and ending of the sentence, better to use "cloud microphysical processes"

2. Line 24: "atmospheric models", may be better to use the "global climate model" instead since some ideal model could describe the evolution of each cloud particle.

3. Line 26: "From the outset, the development of cloud microphysics schemes has resulted in two distinct categories: bulk microphysics parameterization and spectral (bin) microphysics"
   The recently developed Lagrangian particle-based scheme is another type.

4. Line 28: "The spectral (bin) approach represents" added explicitly before represents

5. Line 33: "In climate models with bulk cloud microphysics scheme,"
   may change to "In bulk cloud microphysics schemes of climate models"

6. Line 65: "number density"    actually, it is number mixing ratio

7. Line 67: "number concentration" →number mixing ratio

8. Line 127: "30 vertical layers", the default model setting is 32 layers, did you reset it to 30 layers?

9. Line 135: "the PSD of ICs and $\mu_i$-related cloud microphysical processes are first illustrated by off-line tests."

   PSD is illustrated by off-line tests looks wired, may try: "the impacts of $\mu$ on …. are investigate using off-line test"

10. Line 139: "Fig. 1 shows the impact of $\mu_i$ on the PSDs." --> Fig. 1 shows the impact of $\mu$ on the normalized PSD of cloud ice.

11. Line 141-144: using normalized PSD instead of PSD for accuracy.

12. Line 143: "in terms of number" is it number fraction or number?

13. Line 145: "large IC scenario" large size or mass?

14. Line 191: "climate equilibrium states." We usually use "equilibrium climate states".

15. Line 194: "cloud microphysical process during one model time step." Looks weird, is it process rate?

16. Line 211: "in the tropopause region, where homogeneous freezing produces a large number of ICs (not shown) due to sufficient soluble aerosol particles," in the tropics, is a large number of ICs from convection detrainment and homogeneous freezing of cloud droplet or from "sufficient soluble aerosol particles"?

17. Line 355: "increases atmospheric stability via the radiative budget and then leads to weaker convective precipitation" How? could the author give a more detailed explanation?

**Technical Comments:**

1. Line 9-12: "The calculating formulas of statistical 10 mean radii indicate that, under the same mass (qi) and number (Ni) of ice crystals, the ratios of the mass-weighted radius (Rqi, not related to $\mu_i$) to other statistical mean radii (e.g., effective radiative radius) are completely determined by $\mu_i$."

   This sentence is too long, separating it into two or three sentences.

2. Line 22: "cloud-related process", to be consistent with the beginning and ending of

the sentence, better to use "cloud microphysical processes"

3. Eq. (1), (2), (3)… according to GMD publication format, a comma should be added after equations.

4.  Separate Fig. 1 and Fig. 3 into two Figures, currently they are too small.

---

## Author Comment (AC1)

**Reviewer 1**

We thank the reviewer for the constructive suggestions and valuable comments for improving this manuscript. We hope that the modified manuscript and our response to the comments are satisfactory. The reviewer's comments are in italics and our responses in standard font below. Line number in our responses are referring to the revised manuscript.

***General Comments:***

*Global Climate Models (GCMs) simulate ice clouds using "two-moment schemes" having prognostic solutions for ice particle number concentration (Ni) and cloud ice water content (IWC). The ice particle size distribution (PSD) is generally parameterized as a gamma PSD as described in Eq. 1 in this manuscript, consisting of three variables: N0, λ and μ. Knowing Ni and IWC, N0 and λ are solved for, but in order to obtain mathematical closure for the PSD, μ is given an arbitrary constant value. To my knowledge, there are very few papers investigating the impact of μ on ice cloud microphysical processes in climate models, and this paper appears to be the most thorough and appropriate for GCMs to date. The paper is well organized and well written, providing important new findings relevant to a GMD readership.*

Reply: We do appreciate the positive comments.

*In the study by Mitchell et al. (2006, Atmos. Res.), a snow growth model was developed and tested against aircraft PSD probe measurements where the aircraft descends from cloud top to cloud base under quasi-steady state widespread snowfall conditions, where PSDs were modeled as gamma PSDs. Optimal agreement between the height-evolution of measured and predicted PSDs was obtained for a μ value of -0.6. Assuming the snow growth model was developed properly, this suggests that the PSDs sampled on this flight were characterized by slightly negative μ. Other studies (e.g., Herzegh and Hobbs, 1985, QJRMS; Gordon and Marwitz, 1986, JAS; Mitchell, 1988, JAS) suggest μ typically ranges between 1 and -1 in ice clouds, while Heymsfield (2003, JAS, Part 2) finds μ lies mostly between -2 and 2 when natural ice PSDs are parameterized as gamma PSDs. Thus, the μ values of 2 and 5 assumed for ice clouds in this study appear atypical, but the impacts of changing μ from 0 to 2 (shown in this study) are relevant to real cloud microphysical and radiative processes. This paper would be much more realistic and useful if it also*

*evaluated the impact of changing μ from 0 to -1. Negative values of μ are common when ice crystal nucleation rates are relatively high (Herzegh and Hobbs, 1985, QJRMS).*

Reply: Thanks for these comments. We are in complete agreement that it would be more realistic and useful to evaluate the impact of changing $\mu$ from 0 to $-1$.

The three-parameter gamma distributions of the form $N'(D)=N_0D^\mu e^{-\lambda D}$ are usually used to represent the measured particle size distribution (PSD, bin-averaged data). The three parameters (i.e., $N_0$, $\lambda$, $\mu$) are derived using fitting techniques. The PSDs are relatively wide with negative $\mu$ (Fig. R1). Under negative $\mu$, the particle number densities ($N'$) are increased with decreasing $D$, and become very huge at $D < 1$ μm (Fig. R1 left). In the real world, the ice crystals are usually not less than 1 μm. Furthermore, in the study of Heymsfield (2003, JAS, Part 2), the author pointed out that the gamma-fitted PSD partially compensates for the absence of aircraft PSD measurements below 50 μm, although the functional form used for ice particles below 50 μm represents an extrapolation that is not known explicitly from the data. When only considering the ice crystals over sizes (D) from as small as 10 μm to as large as 2000 μm (measured ice crystal size), the uncertainty from the extrapolation below 50 μm is negligible in the linear space of particle size (Fig. R1 right). Generally speaking, the contribution from very small particles is usually neglected for getting the gamma-fitted PSD from observations.

[Figure]

**Figure R1.** The particle number densities ($N'$, which is a function of $D$) calculated from the gamma distributions with various $\mu$ values. Here, the total particle number and mass-weighted diameter ($D_q$) are set to $10^5$ kg$^{-1}$ (~50 L$^{-1}$) and 40 μm, respectively. Note different markers in the Y-Axis and X-Axis between the left column and right column. The vertical black line indicates $D$= 50 μm.

While observations showed negative $\mu$, the bulk cloud microphysics schemes usually constrain $\mu$ to be nonnegative. Eidhammer et al. (2017, JC) explained this by the singularity of gamma distribution ($N'(D)=N_0D^\mu e^{-\lambda D}$). Under negative $\mu$, the particle number density (i.e., $N'$, which is a function of $D$) would be non-finite at $D$=0. Here, we discuss the reason in more detail. In the bulk cloud scheme, the other two gamma distribution parameters ($N_0$ and $\lambda$) are calculated by the particle's mass ($q$) and number

($N$), and some $\mu$'s gamma functions ($\lambda = [\frac{\pi\rho}{6}\frac{N}{q}\frac{\Gamma(4+\mu)}{\Gamma(1+\mu)}]^{1/3}$, $N_0 = \frac{N\lambda^{(1+\mu)}}{\Gamma(1+\mu)}$). Because the negative integer

and zero are the singularity of the gamma function, the $\mu$ must be greater than $-1$ in these calculation

formulas (i.e., $N_0=\ldots$ and $\lambda=\ldots$). Furthermore, because the gamma function, $\Gamma(x) = \int_0^\infty t^{x-1}e^{-t}\,dt$, is

used for derivating these calculation formulas, the $q$ and $N$ in these calculation formulas indicate the

mass and number of particles with radius from 0 to $\infty$ (hereafter, mathematical size range). Under

negative $\mu$ (the $N'$ is very huge at $D < 1$ µm, Fig. R1 left), more attention should be paid to using the

gamma function because it integrates from 0 to $\infty$. Fig. R2 shows the relative number (upper panel) and

mass (lower panel) contributions from each radius bin of ice crystals. At $\mu_i \geq 0$ (i.e., $\mu_i= 0$, 2, and 5), both

the number and mass contributions are mostly in the radius range from 1 µm to 1000 µm (hereafter,

realistic size range). However, at $\mu_i = -0.9$, the number of particles with a radius from 1 µm to 1000 µm

(i.e., realistic size range) only contributes ~1/3 to the total number (i.e., the $N$ from the mathematical size

range). In other words, the calculation formulas for gamma distribution parameters ($N_0$ and $\lambda$) used in

the bulk cloud schemes are not suited for representing realistic cloud particles. Therefore, our study only

evaluates the impacts of changing $\mu_i$ from 0 to 2, 5.

[Figure]

**Figure R2.** The relative number (upper panel) and mass (lower panel) contributions from each radius bin of ice crystals (ICs). Each bin width is the same based on the logarithm of the particle radius. $N_i$ and $q_i$ are the total number and mass of ICs, respectively. A total of 100 bins were used here. The solid lines indicate the normal IC scenario (i.e., $R_{qi} = 20$ µm), and the dotted lines indicate the large IC scenario (i.e., $R_{qi} = 60$ µm).

*Mitchell (1991, JAS) found that for negative μ values (i.e., superexponential PSD), aggregation was the only growth process that substantially increased ice particle sizes, whereas for positive μ, both aggregation and vapor diffusion contributed to ice particle size increases. This is an example of how ice particle growth processes act differently depending on the sign of μ and illustrates the need to consider both positive and negative μ values.*

Reply: Thanks for this comment. It is clear that the PSD of ice crystals (ICs) becomes wider with decreasing μ, and the PSD is very wide at $\mu_i \leq -0.6$ (Fig. R2). Therefore, as compared to positive $\mu_i$ (i.e., relatively narrow PSD), the interaction between small ICs and large ICs (e.g., the accretion of small ICs by large ICs) should become more important under negative $\mu_i$ (i.e., wide PSD). Although the sensitivity experiments with negative $\mu_i$ were not carried out, more discussion about the impact of $\mu_i$ (i.e., PSD is wide or narrow) on ICs growth was added in the revised manuscript (Line 418-420).

**Major Comments:**

*Line 68: Sentence states that μi is not considered in the default MG scheme because μi = 0. While it is true that μi = 0, stating that it was not considered is misleading. On a number of occasions, Hugh Morrison indicated to me that he was seeking more information about μi and was exploring new ways of treating it (prior to the release of CAM5). After consulting with his peers, he decided a value of zero was most reasonable if a fixed value was to be used.*

Reply: Thanks. The "(i.e., not considered)" is removed from the original sentence "…$\mu_i$ is zero (i.e., not considered) in the default MG scheme" (Line 72).

*Table 1: In the MG scheme, the air density prefactor for the mass-weighted ice fall speed is raised to the power of 0.54 (following Heymsfield and Bansemer 2007), not 0.35 as shown in Table 1. The simulations may need to be rerun if this incorrect value of 0.35 was used.*

Reply: Thanks. All default parameters of the CAM6 model (except for the modification of $\mu_i$) were used in this study. In the code, 0.35 is used for ice crystals and 0.54 is used for snows. In the study of Morrison and Gettelman (2008, JC), 0.54 is used for all cloud and precipitation species.

*Line 317: There are references to support this statement; please add some.*

Reply: Thanks. We have added references (Mitchell, 1991, JAS; DeMott et al., 2010, PNAS; Storelvmo et al., 2013, GRL) to support this statement (…the ICs grow faster and their lifetimes become shorter) in the revised manuscript (Line 322).

*Lines 381-383: This finding appears similar to that reported in Mitchell (1991, JAS) titled "Evolution of snow size spectra in cyclonic storms. Part II: Deviations from the exponential form", where it was found that the IC vapor deposition process was accelerated by increasing μi.*

Reply: Thanks. In the revised manuscript, we pointed out "this is consistent with the previous finding (ICs vapor deposition process is obviously accelerated by increasing $\mu_i$) reported by Mitchell (1991)" (Line 168).

**Technical Comments:**

*Line 47: Common Atmosphere Model => Community Atmosphere Model?*

Reply: Thanks. Done.

*Line 89: PDF => PSD?*

Reply: Thanks. Done.

*Caption for Fig. 2 near bottom: "The two black lines" => "The two black dashed curves"?*

Reply: Thanks. Done.

*Lines 279-280: Are the superscripts for \*Mu and Mu correct?*

Reply: Thanks. The $R_{qi}^*$ denotes the updated $R_{qi}$. Here, $R_{qi}^{*Mu0}$ and $R_{qi}^{Mu0}$ indicate the $R_{qi}^*$ and $R_{qi}$ from the Mu0 experiment. In the revised manuscript, to avoid misunderstanding, a comma was added between "\*" and the experiment name (e.g., $R_{qi}^{*,Mu0}$). Furthermore, more introduction about $R_{qi}^*$ was added, such as "The $R_{qi}^*$ denotes the updated $R_{qi}$, which includes the changes caused by the deposition/sublimation and autoconversion processes at this model time step (Line 284).

---

## Author Comment (AC2)

**Reviewer 2**

We thank the reviewer for the time spent to evaluate our study and for the useful comments. We hope that the modified manuscript and our response to the comments are satisfactory. The reviewer's comments are in italics and our responses in standard font below. Line number in our responses are referring to the revised manuscript.

**General Assessment:**

*Zhang et al. investigate the impact of the shape parameter of cloud ice on simulated cloud properties and radiation using Community Atmosphere Model Version 6 (CAM6) of Community Earth System Model Version 2. The shape parameter is one of three parameters of the gamma distribution, which is commonly used in the two-moment bulk cloud microphysics scheme. Considering the current microphysics schemes usually set shape parameter to 0, investigate the impact of different values of shape parameter on the cloud properties providing important findings relevant to a GMD readership.*

*The authors conducted offline analysis and 10-year simulation using different value (0-5) of the shape parameter, in order to show how the shape parameter influences the cloud properties and radiation transfer. The authors suggested that increasing the value of the shape parameter would lead to higher qi and lower Ni in most regions globally, furthermore, the longwave cloud radiative forcing increases by 5.58 W m−2 (25.11%), and the convective precipitation rate decreases by −0.12 mm day−1 (7.64%).*

*After serious consideration, I here recommend this manuscript subject to major revisions.*

Reply: Thanks for the positive comment.

**Major Comments:**

1) *If I understand correctly, the authors only added the shape parameter to the PSD of cloud ice, the PSD of snow is not considered. This should be indicated in the manuscript, because "ice crystal" represents all the ice particles (cloud ice and snow togther).*

Reply: Thanks. In the revised manuscript, we pointed out "ice crystal" only represents cloud ice in this study (Line 42).

2) *Observation data of ice water path (IWP), shortwave cloud forcing (SWCF), and longwave cloud forcing (LWCF) are need in Figure 4, Table 4, and related analysis. This will give important information of which value of shape parameter give more realistic simulation results compared with observation.*

Reply: The following figure and table (Fig. R1 and Table R1) show observations and simulation results. Besides the model simulations used in the manuscript (horizontal resolution is 1.9°×2.5°), we also carried out three more simulations using the same model code but higher resolution (0.9°×1.25°). Note that, the default namelist configuration (except for $\mu_i$) of model code (namelist_defaults_cam.xml) was used in all simulations. Some tunable parameters of the cloud scheme are different under different horizontal resolutions. For instance, the default size threshold ($R_{cs}$) used for the autoconversion of ice crystal to snow has significant impacts on ice clouds (e.g., ice crystal size and ice water content; Eidhammer et al., 2014). Under the same model code (CESM2.1.3), the $R_{cs}$ is set to 100 μm and 250 μm for 1.9°×2.5° and 0.9°×1.25°, respectively. The reference experiment (i.e., the Mu0 experiment) shows that the simulated cloud radiative effects ($CRE_{LW}$ and $CRE_{SW}$) from both low (1.9°×2.5°) and high (0.9°×1.25°) resolutions agree well with the observations. Unlike the cloud radiative effects, the ice water path (IWP) from the reference experiment is obviously less than the observations, especially for the low-resolution simulation. After increasing $\mu_i$ (i.e., changing $\mu_i$ from 0 to 2, 5), the IWP significantly increases and is closer to observation. Meanwhile, both $CRE_{LW}$ and $CRE_{SW}$ become stronger and corresponding model biases might become a little more obvious.

This study only focuses on the impacts of $\mu_i$, the default tunable parameters (except for $\mu_i$) are used in all the simulations. After improving the representation of $\mu_i$-related processes, further model tuning and analyses are required based on the updated cloud scheme. Therefore, this study does not estimate which value of $\mu_i$ could lead to a better simulation. This was clearly mentioned in the revised manuscript (Line 414-416).

The low-resolution and high-resolution sensitivity experiments show similar changes in cloud radiative effects and precipitation rates (ΔMu2 and ΔMu5, Fig. R1 and Table R1). This indicates that the primary mechanisms for $\mu_i$'s impacts on climate simulation are relatively robust, and not affected by $R_{cs}$.

[Figure]

**Figure R1.** Annual zonal mean distributions of the ice water path (IWP), longwave and shortwave cloud radiative effects (CRE_LW and CRE_SW) from observations (Loeb et al., 2018; Li et al., 2020) and our model experiments (Mu0, Mu2, and Mu5). The left panel represents the model simulations used in the manuscript, and the left panel represents the model simulations with relatively high resolution. The horizontal resolution and corresponding Rcs values are shown on the top.

**Table R1.** The global annual mean ice water path (IWP), longwave and shortwave cloud radiative effects (CRE_LW and CRE_SW), and convective and large-scale precipitation rates (RainC and RainL) from model simulations with two different resolutions (1.9°×2.5° and 0.9°×1.25°) and observations. "Δ" is used to denote the difference from the Mu0 experiment. The corresponding standard deviations calculated from the difference of each year for 10 years are shown in brackets.

| Resolutions | 1.9°×2.5° | | | 0.9°×1.25° | | | |
|---|---|---|---|---|---|---|---|
| Experiments | Mu0 | ΔMu2 | ΔMu5 | Mu0 | ΔMu2 | ΔMu5 | OBS |
| IWP (g m$^{-2}$) | 4.74 | 1.37(0.10) | 2.97(0.06) | 12.97 | 10.69(0.21) | 21.77(0.31) | 24.74 |
| CRE$_{LW}$ (W m$^{-2}$) | 22.22 | 3.40(0.12) | 5.58(0.13) | 23.82 | 4.51(0.15) | 7.27(0.17) | 25.78 |
| CRE$_{SW}$ (W m$^{-2}$) | −49.25 | −3.00(0.43) | −5.34(0.31) | −47.79 | −4.38(0.31) | −7.37(0.39) | −45.33 |
| RainC (mm day$^{-1}$) | 1.57 | −0.08(0.01) | −0.12(0.01) | 1.40 | −0.09(0.01) | −0.13(0.01) | |
| RainL (mm day$^{-1}$) | 1.39 | 0.01(0.01) | 0.01(0.01) | 1.53 | 0(0.01) | −0.02(0.01) | |

**References**

Eidhammer, T., Morrison, H., Bansemer, A., Gettelman, A., and Heymsfield, A. J.: Comparison of ice cloud properties simulated by the Community Atmosphere Model (CAM5) with in-situ observations, Atmospheric Chemistry and Physics, 14, 10103-10118, https://doi.org/10.5194/acp-14-10103-2014, 2014.

Loeb, N. G., Doelling, D. R., Wang, H. L., Su, W. Y., Nguyen, C., Corbett, J. G., Liang, L. S., Mitrescu, C., Rose, F. G., and Kato, S.: Clouds and the Earth's Radiant Energy System (CERES) Energy Balanced and Filled (EBAF) Top-of-Atmosphere (TOA) Edition-4.0 Data Product, J. Climate, 31, 895–918, https://doi.org/ 10.1175/JCLI-D-17-

0208.1, 2018.

Li, J. F., Xu, K., Jiang, J. H., Lee, W., Wang, L., Yu, J., Stephens, G., Fetzer, E., and Wang, Y.: An Overview of CMIP5 and CMIP6 Simulated Cloud Ice, Radiation Fields, Surface Wind Stress, Sea Surface Temperatures, and Precipitation Over Tropical and Subtropical Oceans, J. Geophys. Res.-Atmos., 125, e2020JD032848, https://doi.org/10.1029/2020jd032848, 2020.

3) *From Figure 4, the IWP, LWCF, and SWCF have dramatic changes over the tropics, with much smaller changes in the mid-and high latitude. As we know most of the cloud and precipitation process over the tropics is governed by the convection scheme. Since the shape parameter is only introduced in the large-scale microphysics, while the convection scheme is not changed. Why do the IWP, precipitation, and cloud forcing have the largest change over the tropics? The reader may expect that the largest change is notiecd in the mid-and high latitude, where larger-scale microphysics treated most of the cloud process, therefore, your modifications should have a larger impact over there. Considering the convection precipitation changed also largely, this may indicate the change of IWP, precipitation, and cloud forcing is directly caused by the convection process (as a result of changed climate state), not by the shape parameter in the microphysics scheme.*

Reply: The treatment of clouds in climate models is usually divided into two categories: convective cloud scheme with simplified cloud microphysics and large-scale stratiform cloud scheme with relatively detailed cloud microphysics (e.g., considering the cloud particle size distribution). The intense convective activity over the tropics is usually very short (a few model time steps). The detrained water and ice from convective activity (e.g., cirrus anvils) could left in the atmosphere for a relatively longer time. The convective detainment is usually considered as a source of stratiform clouds and would be treated by the stratiform cloud scheme. Note that, the main cloud formation mechanism is the condensation/deposition of cloud water and ice calculated from the stratiform cloud scheme. More information about the representation of clouds can be found in the model description. The single column model version is a good tool for illustrating the simulated cloud evolution.

Although the tropical precipitation is mainly from the convective scheme, the cloud over the tropics is mostly represented by the larger-scale stratiform cloud scheme. Furthermore, the cloud ice mass mixing ratio ($q_i$) is obviously larger in the upper tropical troposphere (Fig. 2 in the manuscript). Therefore, the modifications in the stratiform cloud scheme have a larger impact over the tropics.

*Minor Comments:*

1) *Line 22: "cloud-related process", to be consistent with the beginning and ending of the sentence, better to use "cloud microphysical processes"*

Reply: Thanks. "cloud-related processes" not only includes "cloud microphysical processes" but also "diagnosing cloud optical properties".

2) *Line 24: "atmospheric models", may be better to use the "global climate model" instead since some ideal model could describe the evolution of each cloud particle.*

Reply: Thanks. "atmospheric models" not only includes "global climate model" but also some other kinds of atmospheric models (e.g., regional climate model and cloud-resolving model). In the revised manuscript, we added "commonly-used" before "atmospheric models" (Line 23 and 26).

3) *Line 26: "From the outset, the development of cloud microphysics schemes has resulted in two distinct categories: bulk microphysics parameterization and spectral (bin) microphysics"*
   *The recently developed Lagrangian particle-based scheme is another type.*

Reply: Thanks. We pointed out that the atmospheric models exclude the ideal model with the recently developed Lagrangian particle-based scheme (Line 26).

4) *Line 28: "The spectral (bin) approach represents" added explicitly before represents*

Reply: Thanks. Done (Line 30).

5) *Line 33: "In climate models with bulk cloud microphysics scheme," may change to "In bulk cloud microphysics schemes of climate models"*

Reply: Thanks. "In climate models with bulk cloud microphysics scheme" was changed to "In the bulk cloud microphysics schemes used for climate models" (Line 36).

6) *Line 65: "number density" actually, it is number mixing ratio*

Reply: Here, "number density" indicates $\delta N/\delta D$. The $N$ and $D$ are ice crystal number and size, respectively.

7) *Line 67: "number concentration"    actually, number mixing ratio*

Reply: Thanks. The original sentence was rewritten to "…in-cloud IC mass and number mixing ratio ($q_i$ and $N_i$, prognostic variables in units of kg kg$^{-1}$ and kg$^{-1}$, respectively)" (Line 70).

8) *Line 127: "30 vertical layers", the default model setting is 32 layers, did you reset it to 30 layers?*

Reply: Thanks. The default configuration and tunable parameters (except for $\mu_i$) were used for all the simulations of this study. The vertical layer is 32.

9) *Line 135: "the PSD of ICs and μi-related cloud microphysical processes are first illustrated by off-line tests." PSD is illustrated by off-line tests looks wired, may try: "the impacts of μi on .... are investigate using off-line test"*

Reply: Thanks. Done (Line 138).

10) *Line 139: "Fig. 1 shows the impact of μi on the PSDs." --> Fig. 1 shows the impact of μ on the normalized PSD of cloud ice.*

Reply: Thanks. Done (Line 142).

11) *Line 141-144: using normalized PSD instead of PSD for accuracy.*

Reply: The shape of PSD (i.e., the relative number or mass contributions of each bin) is the normalized PSD.

12) *Line 143: "in terms of number" is it number fraction or number?*

Reply: Because the normalized PSD is diagnosed by the relative number or mass contributions of each size bin, the "number" indicates "number fraction".

13) *Line 145: "large IC scenario" large size or mass?*

Reply: The large ice crystal (IC) scenario indicates the mass-weighted radius of ICs is 60 μm (i.e., $R_{qi}$ = 60 μm).

*14) Line 191: "climate equilibrium states." We usually use "equilibrium climate states".*

Reply: Thanks. Done (Line 195).

*15) Line 194: "cloud microphysical process during one model time step." Looks weird, is it process rate?*

Reply: "δ" is used to denote the changes in cloud properties that are caused by the cloud microphysical process during one model time step (tendency × one time step).

*16) Line 211: "in the tropopause region, where homogeneous freezing produces a large number of ICs (not shown) due to sufficient soluble aerosol particles," in the tropics, is a large number of ICs from convection detrainment and homogeneous freezing of cloud droplet or from "sufficient soluble aerosol particles"?*

Reply: Homogeneous freezing of ambient liquid aerosols produces a large number of ICs.

*17) Line 355: "increases atmospheric stability via the radiative budget and then leads to weaker convective precipitation" How? could the author give a more detailed explanation?*

Reply: As compared to warm cloud ($CRE_{LW}$ is weaker than $CRE_{SW}$), ice cloud absorbs the earth's surface outgoing longwave radiation more efficiently ($CRE_{LW}$ is stronger than $CRE_{SW}$), and then exerts a net warming radiative effect (Boucher et al., 2013). As shown in Andrews et al. (2010), the fast response in precipitation is strongly correlated with the atmospheric component of changes in radiative fluxes, which is defined as changes in the atmospheric absorption and is calculated as the difference between changes in net TOA radiative fluxes and changes in net surface radiative fluxes. Ice clouds increasing can heat the tropopause, enhance atmospheric stability, and then inhibit convective activity to some extent (Wang et al., 2014).

**References**

Andrews, T., Forster, P. M., Boucher, O., Bellouin, N., and Jones, A.: Precipitation, radiative forcing and global temperature change, Geophysical Research Letters, 37, L14701, https://doi.org/10.1029/2010GL043991, 2010.

Boucher, O., D. Randall, P. Artaxo, C. Bretherton, G. Feingold, P. Forster, V.-M. Kerminen, Y. Kondo, H. Liao, U. Lohmann, P. Rasch, S.K. Satheesh, S. Sherwood, B. Stevens and X.Y. Zhang, 2013: Clouds and Aerosols. In: Climate Change 2013: The Physical Science Basis. Contribution of Working Group I to the Fifth Assessment Report of the Intergovernmental Panel on Climate Change edited by: Stocker, T. F., Qin, D., Plattner, G.-K., Tignor, M.,

Allen, S. K., Boschung, J., Nauels, A., Xia, Y., Bex, V., and Midgley, P. M., Cambridge University Press, Cambridge, United Kingdom and New York, USA, 2013.

Wang, M., Liu, X., Zhang, K., and Comstock, J. M.: Aerosol effects on cirrus through ice nucleation in the Community Atmosphere Model CAM5 with a statistical cirrus scheme, Journal of Advances in Modeling Earth Systems, 6, 756-776, https://doi.org/10.1002/2014MS000339, 2014.

**Technical comments:**

1) *Line 9-12: "The calculating formulas of statistical 10 mean radii indicate that, under the same mass (qi) and number (Ni) of ice crystals, the ratios of the mass-weighted radius (Rqi, not related to μi) to other statistical mean radii (e.g., effective radiative radius) are completely determined by μi."*
   *This sentence is too long, separating it into two or three sentences.*

Reply: Thanks. The long sentence was rewritten as "The $\mu_i$'s impact on the statistical mean radii of ice crystals can be analysed based on their calculating formulas. Under the same mass ($q_i$) and number ($N_i$), the ratios of the mass-weighted radius ($R_{qi}$, not related to $\mu_i$) to other statistical mean radii (e.g., effective radiative radius) are completely determined by $\mu_i$" (Line 10-13).

2) *Line 22: "cloud-related process", to be consistent with the beginning and ending of the sentence, better to use "cloud microphysical processes"*

Reply: "cloud-related processes" not only includes "cloud microphysical processes" but also "diagnosing cloud optical properties".

3) *Eq. (1), (2), (3)... according to GMD publication format, a comma should be added after equations.*

Reply: Thanks. We checked this kind of format. "Eq. (a) and (b)" was changed to "Eqs. (a), (b)" and "Eq. (a-c)" was changed to "Eqs. (a), (b), (c)".

4) *Separate Fig. 1 and Fig. 3 into two Figures, currently they are too small.*

Reply: During the evolution of stratiform clouds, the properties of ice clouds (e.g., $q_i$, $N_i$, and $R_{ni}$, including mixed-phase clouds) largely determine the ice-phase cloud microphysical processes. Meanwhile, these cloud microphysical processes in turn change the cloud properties. They interact as both cause and effect and finally reach climate equilibrium states. To facilitate the subsequent analyses, the cloud properties

and $\mu_i$-related cloud microphysical processes are shown together in one figure. This study only focuses on illustrating the main mechanisms for $\mu_i$'s impacts based on annual zonal mean in-cloud variables. These figures with many variables (Fig. 2 and 3) are acceptable.

---

## Referee Report (RR1)

Second Review for Geoscientific Model Development (GMD)
29 August 2022

Title: Impacts of Ice-Particle Size Distribution Shape Parameter on Climate Simulations with the Community Atmosphere Model Version 6 (CAM6)
Author(s): Wentao Zhang, Xiangjun Shi1, and Chunsong Lu
MS No.: gmd-2022-95
MS type: Model evaluation paper

Reviewer 1 response to the author's response concerning General Comments of 1st review:

The authors have done an outstanding job in replying to the review comments under General Comments.  I recommend that a portion of this analysis provided by the authors be included in the final publication of their manuscript.

First, it's good news that, as the authors state, "We are in complete agreement that it would be more realistic and useful to evaluate the impact of changing $\mu$ from 0 to −1."  Given this, there should be some discussion about this in the revised manuscript, providing a rationale for limiting the study to positive (or zero) $\mu$ values.

The analysis provided in the author's response reveals a limitation in the use of gamma functions for representing ice cloud PSDs in climate and cloud-resolving models.  I suspect most investigators are not aware of these limitations, and therefore recommend that Fig. R1 and an updated Fig. 1 (similar to Fig. R2) be included in the manuscript.  However, I recommend using $\mu$ = -0.5 rather than $\mu$ = -0.6 (used in Fig. R1 and R2) to show that $\mu$ = -0.5 might be a reasonable lower limit for employing gamma functions to realistically approximate PSDs in ice clouds.  That is, $\mu$ = -0.6 in Fig. R2 (left panel) indicates most of the ice particle number concentration $N_i$ is associated with ice particles having a radius > 1 micron, and this would be even more true for $\mu$ = -0.5.  Identifying this lower limit for $\mu$ should be very useful information for investigators that parameterize ice PSD using gamma functions.  As the authors noted, $\mu$ = -0.9 is unrealistic since it predicts ~ 2/3 of the ice particles are having sizes < ~ 1 micron, and such particles are generally counted as aerosol (not as ice crystals).

I have no further comments as the authors have satisfactorily addressed all of my other comments.

---

## Author Response (AR2)

**Reviewer 1**

We thank the reviewer for the valuable suggestion. The reviewer's comments are in italics and our responses are in standard font below. Line number in our responses are referring to the revised manuscript with track changes.

*Comments:*

*The authors have done an outstanding job in replying to the review comments under General Comments. I recommend that a portion of this analysis provided by the authors be included in the final publication of their manuscript.*

*First, it's good news that, as the authors state, "We are in complete agreement that it would be more realistic and useful to evaluate the impact of changing μ from 0 to −1." Given this, there should be some discussion about this in the revised manuscript, providing a rationale for limiting the study to positive (or zero) μ values.*

*The analysis provided in the author's response reveals a limitation in the use of gamma functions for representing ice cloud PSDs in climate and cloud-resolving models. I suspect most investigators are not aware of these limitations, and therefore recommend that Fig. R1 and an updated Fig. 1 (similar to Fig. R2) be included in the manuscript. However, I recommend using μ = -0.5 rather than μ = -0.6 (used in Fig. R1 and R2) to show that μ = -0.5 might be a reasonable lower limit for employing gamma functions to realistically approximate PSDs in ice clouds. That is, μ = -0.6 in Fig. R2 (left panel) indicates most of the ice particle number concentration Ni is associated with ice particles having a radius > 1 micron, and this would be even more true for μ = -0.5. Identifying this lower limit for μ should be very useful information for investigators that parameterize ice PSD using gamma functions. As the authors noted, μ = -0.9 is unrealistic since it predicts ~ 2/3 of the ice particles are having sizes < ~ 1 micron, and such particles are generally counted as aerosol (not as ice crystals).*

*I have no further comments as the authors have satisfactorily addressed all of my other comments*

Reply: We do appreciate the positive comments. In the revised manuscript, the limitation in the use of gamma functions for representing ice-phase PSDs was discussed in the Appendix. The $\mu = -0.6$ (used in

Fig. R1 and R2) was replaced by $\mu = -0.5$. Furthermore, a table was added, which lists the contributions of ice crystals (ICs) from the realistic size range (i.e., 1~1000 μm) to the total number (i.e., the $N_i$ with sizes from 0 to $\infty$) under different mass-weighted radius ($R_{qi}$) scenarios ($R_{qi} = 5, 20, 60$ μm). At $\mu_i = -0.5$, the contributions of ICs from the realistic size range can reach 2/3 except for the small ICs scenario (i.e., $R_{qi} = 5$ μm).